# POLY-VIEW CONTRASTIVE LEARNING

**Amitis Shidani**[*]
Department of Statistics
University of Oxford, UK
shidani@stats.ox.ac.uk

**Devon Hjelm, Jason Ramapuram, Russ Webb,
Eeshan Gunesh Dhekane, and Dan Busbridge**
Apple
dbusbridge@apple.com

## ABSTRACT

Contrastive learning typically matches *pairs* of related views among a number of unrelated negative views. Views can be generated (e.g. by augmentations) or be observed. We investigate matching when there are *more than two* related views which we call *poly-view tasks*, and derive new representation learning objectives using information maximization and sufficient statistics. We show that with unlimited computation, one should maximize the number of related views, and with a fixed compute budget, it is beneficial to decrease the number of unique samples whilst increasing the number of views of those samples. In particular, poly-view contrastive models trained for 128 epochs with batch size 256 outperform Sim-CLR trained for 1024 epochs at batch size 4096 on ImageNet1k, challenging the belief that contrastive models require large batch sizes and many training epochs.

## 1 INTRODUCTION

Self-Supervised Learning (SSL) trains models to solve *tasks* designed take advantage of the structure and relationships within unlabeled data (Bengio et al., 2013; Balestriero et al., 2023; Logeswaran & Lee, 2018; Baevski et al., 2020; Grill et al., 2020). Contrastive learning is one form of SSL that learns representations by maximizing the similarity between conditionally sampled views of a single data instance (positives) and minimizing the similarity between independently sampled views of other data instances (negatives) (Qi & Su, 2017; van den Oord et al., 2018; Bachman et al., 2019; Hénaff et al., 2019; He et al., 2019; Tian et al., 2020a;b; Chen et al., 2020a).

One principle behind contrastive learning is Mutual Information (MI) maximization (van den Oord et al., 2018; Hjelm et al., 2019). Many works have elucidated the relationship between contrastive learning and information theory (Poole et al., 2019; Tschannen et al., 2020; Lee et al., 2023; Gálvez et al., 2023). However, MI maximization is only part of the story (Tschannen et al., 2020); successful contrastive algorithms rely on negative sampling (Wang & Isola, 2020; Robinson et al., 2021; Song et al., 2016; Sohn, 2016) and data augmentation (Bachman et al., 2019; Tian et al., 2020b; Chen et al., 2020a; Fort et al., 2021; Balestriero et al., 2022b;a) to achieve strong performance.

While it is possible to design tasks that draw any number of views, contrastive works typically solve *pairwise tasks*, i.e. they maximize the similarity of *exactly two* views, or *positive pairs* (Balestriero et al., 2023; Tian et al., 2020a). The effect of more views, or increased *view multiplicity* (Bachman et al., 2019), was investigated in SSL (van den Oord et al., 2018; Hjelm et al., 2019; Tian et al., 2020a; Caron et al., 2020). However, these works optimize a linear combination of pairwise tasks; increasing view multiplicity mainly improves the gradient signal to noise ratio of an equivalent lower view multiplicity task, as was observed in supervised learning (Hoffer et al., 2019; Fort et al., 2021).

In this work, we investigate increasing view multiplicity in contrastive learning and the design of SSL tasks that use many views. We call these tasks *poly-view* to distinguish them from *multi-view*, as *multi* usually means *exactly two* (Tian et al., 2020a; Balestriero et al., 2023). In addition to improved signal to noise (Hoffer et al., 2019; Fort et al., 2021), poly-view tasks allow a model to access many related views at once, increasing the total information about the problem. We show theoretically and empirically that this has a positive impact on learning. We make the following contributions:

1. We generalize the information-theoretic foundation of existing contrastive tasks to poly-view (Section 2.3), resulting in a new family of representation learning algorithms.

---

[*]Work done during an internship at Apple. For a detailed breakdown of author contributions see Appendix I.

2. We use the framework of sufficient statistics to provide an additional perspective on contrastive representation learning in the presence of multiple views, and show that in the case of two views, this reduces to the well-known SimCLR loss, providing a new interpretation of contrastive learning (Section 2.4) and another new family of representation learning objectives.

3. Finally, we demonstrate poly-view contrastive learning is useful for image representation learning. We show that higher view multiplicity enables a new compute Pareto front for contrastive learning, where it is beneficial to reduce the batch size and increase multiplicity (Section 3.2). This front shows that poly-view contrastive models trained for 128 epochs with batch size 256 outperforms SimCLR trained for 1024 epochs at batch size 4096 on ImageNet1k.

## 2 VIEW MULTIPLICITY IN CONTRASTIVE LEARNING

We seek to understand the role of *view multiplicity* in contrastive learning (Definition 2.1).

**Definition 2.1** (View Multiplicity). *The view multiplicity $M$ is the number of views per sample. In batched sampling, drawing $K$ samples results in $V = M \times K$ views per batch. (Hoffer et al., 2019).*

Multiple data views may occur naturally as in CLIP (Radford et al., 2021) or, as is our primary interest, be samples from an augmentation policy as is common in SSL. Our goal is to develop tasks

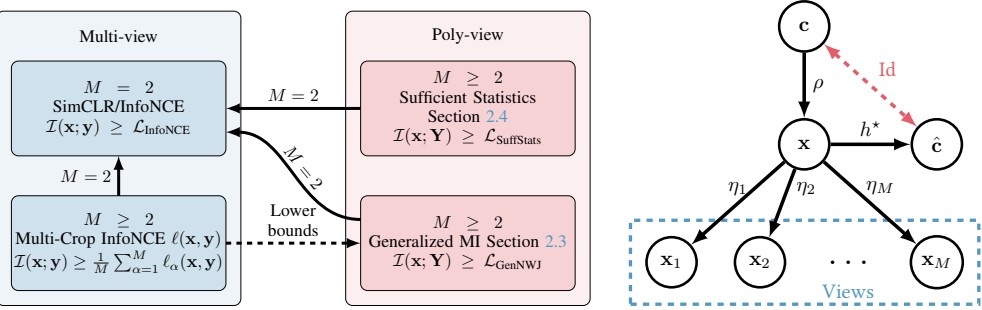

(a) View multiplicity in contrastive learning.  (b) View multiplicity generative process.

Figure 1: *(a)* The role of multiplicity in contrastive learning. $\mathcal{I}(\mathbf{x}; \mathbf{y})$ present the MI between two random variables $\mathbf{x}$ and $\mathbf{y}$, while $\mathcal{I}(\mathbf{x}; \mathbf{Y})$ is the MI between $\mathbf{x}$ and the set of Random Variable (RV)s $\mathbf{Y}$. $\mathcal{L}_{\text{Method}}$ denotes the contrastive lower-bound achieved by each method, ignoring the constants. In the multi-crop box, $\ell_\alpha(\mathbf{x}, \mathbf{y})$ is the contrastive lower-bound produced by the $\alpha$-th crop/view. *(b)* The multiple view sample generation with generative factor $\mathbf{c}$, where the main sample is generated through the generative process $\rho$, and views are generated through different view-generation processes $\eta_\alpha$ for $\alpha \in [M]$, e.g. augmentations. The goal is to find the map $h^\star$ such that the reconstructed generative factor $\hat{\mathbf{c}}$ recovers $\mathbf{c}$, hence the identity map.

that can use multiplicity $M$. We start by presenting the generative process underlying multiplicity (Section 2.1). We then consider optimizing many pairwise tasks (Section 2.2), known as Multi-Crop, and show that Multi-Crop *reduces the variance* of the corresponding paired objective but *cannot* improve bounds on quantities like MI. Next, we revisit the information theoretic origin of InfoNCE, and derive new objectives that solve tasks across *all views* and *do not* decompose into pairwise tasks (Section 2.3). Finally, as the framework of sufficient statistics is natural at high multiplicity, we use it to derive new objectives which solve tasks across *all views* (Section 2.4). All of these objectives are related, as is shown in Figure 1a. Before proceeding, we introduce our notation.

**Notation**   We denote vector and set of random variables (RVs) as $\mathbf{x}$ and $\mathbf{X}$, with corresponding densities $p_{\mathbf{x}}$ and $p_{\mathbf{X}}$, and realizations $\boldsymbol{x}$ and $\boldsymbol{X}$. Vector realizations $\boldsymbol{x}$ live in spaces denoted by $\mathcal{X}$. The conditional distribution of $\mathbf{y}$ given a realization $\boldsymbol{x}$ is denoted $p_{\mathbf{y}|\mathbf{x}=\boldsymbol{x}}$. The expectation of a scalar function $f : \mathcal{X} \mapsto \mathbb{R}$ is $\mathbb{E}[f(\mathbf{x})] = \mathbb{E}_{\boldsymbol{x} \sim p_{\mathbf{x}}}[f(\boldsymbol{x})]$. For $a \le c \le b$, $\mathbf{X}_{a:b} = \{\mathbf{x}_a, \mathbf{x}_{a+1}, \dots, \mathbf{x}_b\}$ represents a set of RVs, and $\mathbf{X}_{a:b}^{(\neq c)} = \mathbf{X}_{a:b} \setminus \{\mathbf{x}_c\}$. The density of $\mathbf{X}_{a:b}$ is the joint of its constituent RVs. MI between $\mathbf{x}$ and $\mathbf{y}$ is denoted $\mathcal{I}(\mathbf{x}; \mathbf{y})$ and is defined over RV sets as $\mathcal{I}(\mathbf{X}; \mathbf{Y})$. We denote the Shannon and differential entropy of $\mathbf{x}$ as H$(\mathbf{x})$, and the Kullback-Leibler Divergence (KLD) between densities $p$ and $q$ by $\mathcal{D}_{\text{KL}}\left(p \parallel q\right)$. Finally, we write the integer set $\{1, \dots, K\}$ as $[K]$, and use Latin and Greek alphabet to index samples and views respectively.

## 2.1 GENERATIVE PROCESS AND INFOMAX FOR VIEW MULTIPLICITY

We present the causal graph underlying $M$ view $\mathbf{X}_{1:M} = \{\mathbf{x}_\alpha \; ; \; \alpha \in [M]\}$ generation in Figure 1b.

The InfoMax principle (Linsker, 1988) proposes to reconstruct an unknown $\mathbf{c}$ by optimizing $h^\star = \arg\max_{h \in \mathcal{H}} \mathcal{I}(\mathbf{x}, h(\mathbf{x}))$. To avoid trivial solutions, two-view contrastive methods (van den Oord et al., 2018; Hjelm et al., 2019; Hénaff et al., 2019; Tian et al., 2020a) perform InfoMax through a *proxy task* that instead maximizes a lower bound on the MI between two views $\mathcal{I}(h(\mathbf{x}_1); h(\mathbf{x}_2))$. These methods rely on information about $\mathbf{c}$ being in the information shared between each pair of views. A natural extension to two-view contrastive learning is to consider many views, where the total amount of information about $\mathbf{c}$ is potentially larger. In Sections 2.2 to 2.4, we investigate different approaches to solving this generalized InfoMax, beginning with Multi-Crop (Section 2.2) before considering more general MI approaches (Section 2.3) and sufficient statistics (Section 2.4).

## 2.2 LINEAR COMBINATIONS OF PAIR-WISE TASKS

The first approach combines objectives on pairs $\mathbf{x}_\alpha, \mathbf{x}_\beta$ from the set of $M$ views $\mathbf{X}_{1:M}$

$$\mathcal{L}_{\text{Multi-Crop}}(\mathbf{X}_{1:M}) = \frac{1}{M(M-1)} \sum_{\alpha=1}^{M} \sum_{\beta \neq \alpha}^{M} \mathcal{L}_{\text{Pair}}(\mathbf{x}_\alpha, \mathbf{x}_\beta). \tag{1}$$

The objective Equation 1 is the *all-pairs* formulation of Tian et al. (2020a), and corresponds to Multi-Crop (Caron et al., 2020; 2021) in the presence of $M$ global views[1]. For convenience, we will refer to the objective Equation 1 as Multi-Crop. Multi-Crop has been used numerous times in SSL, here we will show how it achieves improved model performance through its connection to InfoMax.

**Proposition 2.1.** *For $K$ independent samples and multiplicity $M$ denoted $\mathbf{X}_{1:K,1:M}$, the Multi-Crop of any $\mathcal{L}_{Pair}$ in Equation 1 has the same MI lower bound as the corresponding $\mathcal{L}_{Pair}$:*

$$\mathcal{I}(\mathbf{x}_1; \mathbf{x}_2) \geq \log(K) - \mathbb{E}\left[\mathcal{L}_{Multi\text{-}Crop}(\mathbf{X}_{1:K,1:M})\right] = \log(K) - \mathbb{E}\left[\mathcal{L}_{Pair}(\mathbf{X}_{1:K,1:2})\right], \tag{2}$$

*where the expectation is over $K$ independent samples (see Appendix C.1 for the proof).*

Proposition 2.1 shows that increasing view multiplicity in Multi-Crop *does not* improve the MI lower-bound compared to vanilla InfoNCE with two views. However, Multi-Crop *does* improve the variance of the MI estimate (Proposition 2.2).

**Proposition 2.2.** *For $K$ independent samples and multiplicity $M$, $M \geq 3$, denoted $\mathbf{X}_{1:K,1:M}$, the Multi-Crop of any $\mathcal{L}_{Pair}$ in Equation 1 has a lower sample variance than the corresponding $\mathcal{L}_{Pair}$:*

$$\text{Var}\left[\mathcal{L}_{Multi\text{-}Crop}(\mathbf{X}_{1:M})\right] \leq \frac{2(2M-1)}{3M(M-1)} \text{Var}\left[L_{Pair}(\mathbf{x}_1, \mathbf{x}_2)\right] < \text{Var}\left[L_{Pair}(\mathbf{x}_1, \mathbf{x}_2)\right], \tag{3}$$

*where the variance is over $K$ independent samples (see Appendix C.2 for the proof).*

Propositions 2.1 and 2.2 show that better Multi-Crop performance follows from improved gradient signal to noise ratio as in the supervised case (Fort et al., 2021) and supports the observations of Balestriero et al. (2022b). See Appendix D for further discussion about Multi-Crop.

## 2.3 GENERALIZED INFORMATION MAXIMIZATION AS CONTRASTIVE LEARNING

In this subsection, we develop our first objectives that use $M$ views at once and do not decompose into objectives over pairs of views as in Section 2.2.

### 2.3.1 GENERALIZED MUTUAL INFORMATION BETWEEN $M$ VIEWS

As InfoNCE optimizes a lower bound on of the MI between two views (van den Oord et al., 2018; Poole et al., 2019), consider the One-vs-Rest MI (Definition 2.2).

---

[1]The original Multi-Crop also takes a mixture of smaller views and compares them to larger views, resulting in a more complicated augmentation policy. As our work is focused on studying the effect of multiplicity, we do not investigate the extra benefits obtainable by also changing the augmentation policy. For investigations into augmentations, see Tian et al. (2020b).

**Definition 2.2** (One-vs-Rest MI). *The One-vs-Rest MI for any $\alpha \in [M]$ given a set of $M \geq 2$ Random Variables (RVs) $\mathbf{X}_{1:M} = \{\mathbf{x}_\alpha \; ; \; \alpha \in [M]\}$ is*

$$\mathcal{I}\left(\mathbf{x}_\alpha; \mathbf{X}_{1:M}^{\neq\alpha}\right) = \mathcal{D}_{\text{KL}}\left(p_{\mathbf{X}_{1:M}} \parallel p_{\mathbf{x}_\alpha} p_{\mathbf{X}_{1:M}^{\neq\alpha}}\right). \tag{4}$$

One-vs-Rest MI (Definition 2.2) aligns with generalized InfoMax (Section 2.1); the larger set $\mathbf{X}_{1:M}^{\neq\alpha}$ can contain more information about the generative factor $\mathbf{c}$. Note that due to the data processing inequality $\mathcal{I}(\mathbf{x}_\alpha; \mathbf{X}_{1:M}^{\neq\alpha}) \leq \mathcal{I}(\mathbf{x}_\alpha; \mathbf{c})$, estimating One-vs-Rest MI gives us a lower-bound on InfoMax.

**Estimating One-vs-Rest MI** Contrastive learning estimates a lower-bound to the MI using a sample-based estimator, for example InfoNCE (van den Oord et al., 2018; Poole et al., 2019) and $\mathcal{I}_{\text{NWJ}}$ (Hjelm et al., 2019; Nguyen et al., 2008). Theorem 2.1 generalizes the $\mathcal{I}_{\text{NWJ}}$ lower-bound for the One-vs-Rest MI (see Appendix C.3 for the proof).

**Theorem 2.1** (Generalized $\mathcal{I}_{\text{NWJ}}$). *For any $M \geq 2$, $\alpha \in [M]$, a set of $M$ random variables $\mathbf{X}_{1:M}$, and for any positive function $F^{(M)} : \mathcal{X} \times \mathcal{X}^{M-1} \mapsto \mathbb{R}^+$*

$$\mathcal{I}(\mathbf{x}_\alpha; \mathbf{X}_{1:M}^{\neq\alpha}) \geq \mathbb{E}_{p_{\mathbf{x}_{1:M}}}\left[F^{(M)}(\boldsymbol{x}_\alpha, \boldsymbol{X}_{1:M}^{\neq\alpha})\right] - \mathbb{E}_{p_{\mathbf{x}_\alpha} p_{\mathbf{X}_{1:M}^{\neq\alpha}}}\left[e^{F^{(M)}(\boldsymbol{x}_\alpha, \boldsymbol{X}_{1:M}^{\neq\alpha})}\right] + 1 = \mathcal{I}_{\text{GenNWJ}}. \tag{5}$$

We can use the $\mathcal{I}_{\text{GenNWJ}}$ lower bound (Theorem 2.1) for any function $F^{(M)} : \mathcal{X} \times \mathcal{X}^{M-1} \mapsto \mathbb{R}^+$. In order to efficiently maximize the MI, we want the bound in Equation 5 to be as tight as possible, which we can measure using the *MI Gap* (Definition 2.3).

**Definition 2.3** (MI Gap). *For any $M \geq 2$, $\alpha \in [M]$, a set of $M$ random variables $\mathbf{X}_{1:M}$, and map $g_\alpha^{(M)} : \mathcal{X} \times \mathcal{X}^{M-1} \mapsto \mathbb{R}^+$ of the form*

$$g_\alpha^{(M)}(\boldsymbol{x}_\alpha, \boldsymbol{X}_{1:M}^{\neq\alpha}) = \frac{p_{\boldsymbol{x}_\alpha} p_{\boldsymbol{X}_{1:M}^{\neq\alpha}}}{p_{\boldsymbol{X}_{1:M}}} e^{F^{(M)}(\boldsymbol{x}_\alpha, \boldsymbol{X}_{1:M}^{\neq\alpha})}, \tag{6}$$

*the MI Gap $\mathcal{G}_{\text{MI}}\left(\mathbf{X}_{1:M}; g_\alpha^{(M)}\right)$ is*

$$\mathcal{G}_{\text{MI}}\left(\mathbf{X}_{1:M}; g_\alpha^{(M)}\right) = \mathcal{I}\left(\mathbf{x}_\alpha; \mathbf{X}_{1:M}^{\neq\alpha}\right) - \mathcal{I}_{\text{GenNWJ}} = \mathbb{E}_{p_{\mathbf{x}_{1:M}}}\left[g_\alpha^{(M)} - \log\left(g_\alpha^{(M)}\right) - 1\right], \tag{7}$$

*where we have written $g_\alpha^{(M)}$ instead of $g_\alpha^{(M)}(\boldsymbol{x}_\alpha, \boldsymbol{X}_{1:M}^{\neq\alpha})$ when the arguments are clear.*

The map $g_\alpha^{(M)}$ in Equation 6 aggregates over $M$ views and is called the *aggregation function*.

### 2.3.2 PROPERTIES OF THE AGGREGATION FUNCTION

The choice of $g_\alpha^{(M)}$ is important as it determines the MI Gap (Definition 2.3) at any multiplicity $M$. As we wish to employ $g_\alpha^{(M)}$ to obtain a lower bound on One-vs-Rest MI, it *should* be

1. *Interchangeable*: $\mathcal{I}(\mathbf{x}_\alpha; \mathbf{X}_{1:M}^{\neq\alpha}) = \mathcal{I}(\mathbf{X}_{1:M}^{\neq\alpha}; \mathbf{x}_\alpha) \implies g_\alpha^{(M)}(\boldsymbol{x}_\alpha, \boldsymbol{X}_{1:M}^{\neq\alpha}) = g_\alpha^{(M)}(\boldsymbol{X}_{1:M}^{\neq\alpha}, \boldsymbol{x}_\alpha)$,

2. *Reorderable*: $\mathcal{I}(\mathbf{x}_\alpha; \mathbf{X}_{1:M}^{\neq\alpha}) = \mathcal{I}[\mathbf{x}_\alpha; \Pi(\mathbf{X}_{1:M}^{\neq\alpha})] \implies g_\alpha^{(M)}(\boldsymbol{x}_\alpha, \boldsymbol{X}_{1:M}^{\neq\alpha}) = g_\alpha^{(M)}[\boldsymbol{x}_\alpha, \Pi(\boldsymbol{X}_{1:M}^{\neq\alpha})]$, where $\Pi(\{x_1, \ldots, x_N\}) = \{x_{\Pi_1}, \ldots, x_{\Pi_N}\}$ is a permutation operator, and

3. *Expandable*: $g_\alpha^{(M)}$ can accommodate different sized rest-sets $\mathbf{X}_{1:M}^{\neq\alpha}$, i.e. can expand to any $M$.

We seek non-trivial lower bounds for the One-vs-Rest MI (Equation 5), and to minimize the MI Gap (Equation 7). The Data Processing Inequality (DPI) gives $\mathcal{I}(\mathbf{x}_\alpha; \mathbf{X}_{1:M}^{\neq\alpha}) \geq \mathcal{I}(\mathbf{x}_\alpha; \mathbf{x}_\beta)$ for all $\mathbf{x}_\beta \in \mathbf{X}_{1:M}^{\neq\alpha}$. So, $\mathcal{I}(\mathbf{x}_\alpha; \mathbf{X}_{1:M}^{\neq\alpha}) \geq (M-1)^{-1} \sum_\beta \mathcal{I}(\mathbf{x}_\alpha; \mathbf{x}_\beta)^2$, provides a baseline for the lower-bound for One-vs-Rest MI, leading us to introduce the following requirement:

4. *Valid*: The aggregation function $g_\alpha^{(M)}$ should give a gap that is at most the gap given by the mean of pairwise comparisons with $g_\alpha^{(2)}$

$$\mathcal{G}_{\text{MI}}\left(\mathbf{X}_{1:M}; g_\alpha^{(M)}\right) \leq \frac{1}{M-1} \sum_{\beta \neq \alpha} \mathcal{G}_{\text{MI}}\left(\{\mathbf{x}_\alpha, \mathbf{x}_\beta\}; g_\alpha^{(2)}\right). \tag{8}$$

---

[2]We note that the objective introduced by Tian et al. (2020a) for the multi-view setting is indeed the average lower-bound we present here.

### 2.3.3 POLY-VIEW INFOMAX CONTRASTIVE OBJECTIVES

We now present the first poly-view objectives, corresponding to choices of $F^{(M)}$ and its aggregation function $g_\alpha^{(M)}$ with the properties outlined in Section 2.3.2. For any function $F^{(2)}$, define $F^{(M)}$, and their aggregation functions correspondingly by Equation 6 as following:

$$\text{Arithmetic average:} \qquad F^{(M)}\left(\boldsymbol{x}_\alpha, \boldsymbol{X}_{1:M}^{\neq\alpha}\right) = \log\left(\frac{1}{M-1}\sum_{\beta\neq\alpha} e^{F^{(2)}(\boldsymbol{x}_\alpha,\boldsymbol{x}_\beta)}\right), \qquad (9)$$

$$\text{Geometric average:} \qquad F^{(M)}\left(\boldsymbol{x}_\alpha, \boldsymbol{X}_{1:M}^{\neq\alpha}\right) = \frac{1}{M-1}\sum_{\beta\neq\alpha} F^{(2)}(\boldsymbol{x}_\alpha,\boldsymbol{x}_\beta). \qquad (10)$$

Both functions satisfy the properties in Section 2.3.2 (see Appendix C.4 for proof).

To establish a connection to contrastive losses, we introduce notation for sampling the causal graph in Figure 1b. From the joint distribution $p_{\mathbf{X}_{1:M}}$, we draw $K$ independent samples denoted by:

$$\{\mathbf{X}_{i,1:M}\}_{i=1}^{K} = \{(\mathbf{x}_{i,1}, \ldots, \mathbf{x}_{i,M})\}_{i=1}^{K} = \left\{\{\mathbf{x}_{i,\alpha}\}_{\alpha=1}^{M}\right\}_{i=1}^{K} = \mathbf{X}_{1:K,1:M} \quad \text{i.e. } \mathbf{X}_{i,\alpha} = \mathbf{x}_{i,\alpha}. \quad (11)$$

Evaluating the functions in Equations 9 and 10 in Theorem 2.1 reveals the lower bound on One-vs-Rest MI and the *Poly-view Contrastive Losses* (Theorem 2.2, see Appendix C.5 for the proof).

**Theorem 2.2** (Arithmetic and Geometric PVC lower bound One-vs-Rest MI). *For any $K$, $M \geq 2$, $B = KM$, $\alpha \in [M]$, any scalar function $f : \mathcal{C} \times \mathcal{C} \mapsto \mathbb{R}$, and map $h : \mathcal{X} \mapsto \mathcal{C}$, we have*

$$\mathcal{I}\left(\mathbf{x}_\alpha; \mathbf{X}_{1:M}^{\neq\alpha}\right) \geq c(B, M) + \mathbb{E}\left[\frac{1}{K}\sum_{i=1}^{K}\log\frac{1}{M-1}\sum_{\beta\neq\alpha}\ell_{i,\alpha,\beta}\right] \equiv c(B, M) - \mathcal{L}_{\text{Arithmetic PVC}}, \quad (12)$$

$$\mathcal{I}\left(\mathbf{x}_\alpha; \mathbf{X}_{1:M}^{\neq\alpha}\right) \geq c(B, M) + \mathbb{E}\left[\frac{1}{K}\sum_{i=1}^{K}\frac{1}{M-1}\sum_{\beta\neq\alpha}\log\ell_{i,\alpha,\beta}\right] \equiv c(B, M) - \mathcal{L}_{\text{Geometric PVC}}, \quad (13)$$

*where $c(B, M) = \log(B - M + 1)$, the expectation is over $K$ independent samples $\mathbf{X}_{1:K,1:M}$, and*

$$\ell_{i,\alpha,\beta}\left(\mathbf{X}_{1:K,1:M}\right) = \frac{e^{f(\widetilde{\mathbf{x}}_{i,\alpha},\widetilde{\mathbf{x}}_{i,\beta})}}{e^{f(\widetilde{\mathbf{x}}_{i,\alpha},\widetilde{\mathbf{x}}_{i,\beta})} + \sum_{j\neq i}\sum_{\gamma=1}^{M} e^{f(\widetilde{\mathbf{x}}_{j,\gamma},\widetilde{\mathbf{x}}_{i,\beta})}}, \qquad \widetilde{\mathbf{x}}_{i,\alpha} = h(\mathbf{x}_{i,\alpha}). \quad (14)$$

*We have written $\ell_{i,\alpha,\beta}$ instead of $\ell_{i,\alpha,\beta}\left(\mathbf{X}_{1:K,1:M}\right)$ where the meaning is clear.*

Maximizing lower-bound means maximizing map $h$, leading to $h^\star$ in Figure 1b. In Appendix C.5, we show $F^{(2)}(\tilde{\mathbf{X}}_{i,\alpha}, \boldsymbol{x}_{i,\beta}) = c(B, M) + \log\ell_{i,\alpha,\beta}$, where $\tilde{\mathbf{X}}_{i,\alpha} = \{\mathbf{X}_{j,\beta}\}_{j\neq i,\beta}\bigcup\{\mathbf{x}_{i,\alpha}\}$.

**Tightness of MI Gap** *Valid* property (Equation 8) ensures that the lower-bound for a fixed $M$ has a smaller MI Gap than the average MI Gap of those views. Without loss of generality, taking $\alpha = 1$, a *valid* solution guarantees that the MI Gap for $M > 2$ is smaller than the MI Gap for $M = 2$. The DPI implies that for $N \geq M$ and fixed $\alpha$, $\mathcal{I}\left(\mathbf{x}_\alpha; \mathbf{X}_{1:M}^{\neq\alpha}\right) \leq \mathcal{I}\left(\mathbf{x}_\alpha; \mathbf{X}_{1:N}^{\neq\alpha}\right)$. One would expect the lower-bound to be also increasing, which indeed is the case. In fact, we can prove more; consider that the MI Gap is monotonically non-increasing with respect to $M$[3], i.e. the MI Gap would either become tighter or stay the same as $M$ grows. We show that the aggregation functions by Equations 9 and 10 have this property (Theorem 2.3, see Appendix C.6 for the proof).

**Theorem 2.3.** *For fixed $\alpha$, the MI Gap of Arithmetic and Geometric PVC are monotonically non-increasing with $M$:*

$$\mathcal{G}_{\text{MI}}(\mathbf{X}_{1:M_2}; g_\alpha^{(M_2)}) \leq \mathcal{G}_{\text{MI}}(\mathbf{X}_{1:M_1}; g_\alpha^{(M_1)}) \quad \forall\, M_1 \leq M_2. \quad (15)$$

**Recovering existing methods** Arithmetic and Geometric PVC optimize One-vs-Rest MI. $M = 2$ gives the two-view MI that SimCLR maximizes and the corresponding loss (see Appendix E.2). Additionally, for a choice of $F^{(2)}$, we recover SigLIP (Zhai et al., 2023b), providing an information-theoretic perspective for that class of methods (see Appendix E.3).

---

[3]Note that this guarantees that the lower-bound is increasing with respect to $M$.

## 2.4 FINDING GENERALIZED SUFFICIENT STATISTICS AS CONTRASTIVE LEARNING

Now we develop our second objectives that use $M$ views at once. Using a probabilistic perspective of the causal graph (Figure 1b), we show how to recover the generative factors with sufficient statistics (Section 2.4.1). We then explain how sufficient statistics connects to InfoMax, and derive further poly-view contrastive losses (Section 2.4.2). Finally, we will see that the approaches of MI lower-bound maximization of Section 2.3, and sufficient statistics are connected.

### 2.4.1 REPRESENTATIONS ARE POLY-VIEW SUFFICIENT STATISTICS

To develop an intuition for the utility of sufficient statistics for representation learning, we begin in the simplified setting of an invertible generative process, $h = \rho^{-1}$, and a lossless view generation procedure $\eta_\alpha$: $\mathcal{I}(\mathbf{c}; \eta_\alpha(\mathbf{x})) = \mathcal{I}(\mathbf{c}; \mathbf{x})$. If the function space $\mathcal{H}$ is large enough, then $\exists\, h \in \mathcal{H}$ such that $\hat{\mathbf{c}} = h(\mathbf{x}) = \mathbf{c}$. Using the DPI for invertible functions, we have

$$\max_{h \in \mathcal{H}} \mathcal{I}(\mathbf{x}; h(\mathbf{x})) = \mathcal{I}(\mathbf{x}; \mathbf{c}) = \max_{h \in \mathcal{H}} \mathcal{I}(h(\mathbf{x}); \mathbf{c}). \tag{16}$$

If we let $h^\star = \arg\max_{h \in \mathcal{H}} \mathcal{I}(\mathbf{x}; h(\mathbf{x}))$, then $h^\star(\mathbf{x})$ is a sufficient statistic of $\mathbf{x}$ with respect to $\mathbf{c}$ (see e.g. Cover & Thomas (2006)), and the information maximization here is related to InfoMax.

If we knew the conditional distribution $p_{\mathbf{x}|\mathbf{c}}$, finding the sufficient statistics $T(\mathbf{x})$ of $\mathbf{x}$ with respect to $\mathbf{c}$ gives $T = h^\star$. In general, we *do not know* $p_{\mathbf{x}|\mathbf{c}}$, and generative processes are typically lossy.

Therefore, to make progress and find $h^\star = \arg\max_{h \in \mathcal{H}} \mathcal{I}(\mathbf{x}; h(\mathbf{x}))$ with sufficient statistics, we need to estimate $p_{\mathbf{x}|\mathbf{c}}$. For this purpose, we use *view multiplicity*; we know from DPI that a larger set of views $\mathbf{X}_{1:M}$ may contain more information about $\mathbf{c}$, i.e. $\mathcal{I}(\mathbf{X}_{1:M_2}; \mathbf{c}) \geq \mathcal{I}(\mathbf{X}_{1:M_1}; \mathbf{c})$ for $M_2 \geq M_1$. Our assumptions for finding the sufficient statistics $T_{\mathbf{y}}(\mathbf{x})$ of $\mathbf{x}$ with respect to $\mathbf{y}$ are

1. The poly-view conditional $p_{\mathbf{x}_\alpha | \mathbf{X}_{1:M}^{\neq \alpha}}$ is a better estimate for $p_{\mathbf{x}_\alpha | \mathbf{c}}$ for larger $M$,

2. All views have the same generative factor: $T_{\mathbf{c}}(\mathbf{x}_\alpha) = T_{\mathbf{c}}(\mathbf{x}_\beta)$,

The representations are given by a neural network and are therefore finite-dimensional. It means that the generative factor is assumed to be finite-dimensional. Fisher-Darmois-Koopman-Pitman theorem (Daum, 1986) proves that the conditional distributions $p_{\mathbf{x}_\alpha | \mathbf{X}_{1:M}^{\neq \alpha}}$ and $p_{\mathbf{x}_\alpha | \mathbf{c}}$ are exponential families, i.e. for some functions $r_1, r_2, T$ and reorderable function (Section 2.3.2) $Q$:

$$p_{\mathbf{x}_\alpha | \mathbf{X}_{1:M}^{\neq \alpha}} = r_1(\mathbf{x}_\alpha)\, r_2(\mathbf{X}_{1:M}^{\neq \alpha}) \exp\left( T_{\mathbf{X}_{1:M}^{\neq \alpha}}(\mathbf{x}_\alpha) \cdot Q(\mathbf{X}_{1:M}^{\neq \alpha}) \right), \tag{17}$$

$$p_{\mathbf{x}_\alpha | \mathbf{c}} = r_1^\star(\mathbf{x}_\alpha)\, r_2^\star(\mathbf{c}) \exp\left( T_{\mathbf{c}}(\mathbf{x}_\alpha) \cdot Q^\star(\mathbf{c}) \right). \tag{18}$$

The first assumption says that for any $M$, it is enough to find the sufficient statistics of $\mathbf{x}_\alpha$ with respect to $\mathbf{X}_{1:M}^{\neq \alpha}$ as an estimate for $T_{\mathbf{c}}(\mathbf{x}_\alpha)$. Since the estimation of the true conditional distribution becomes more accurate as $M$ grows,

$$\limsup_{M \to \infty} \| T_{\mathbf{c}}(\mathbf{x}_\alpha) - T_{\mathbf{X}_{1:M}^{\neq \alpha}}(\mathbf{x}_\alpha) \| \to 0, \quad \limsup_{M \to \infty} \| Q^\star(\mathbf{c}) - Q(\mathbf{X}_{1:M}^{\neq \alpha}) \| \to 0. \tag{19}$$

We see that sufficient statistics gives us a new perspective on InfoMax for representation learning: representations for $\mathbf{x}$ *are sufficient statistics* of $\mathbf{x}$ with respect to the generative factor $\mathbf{c}$, which can be approximated by sufficient statistics of one view $\mathbf{x}_\alpha$ with respect to the other views $\mathbf{X}_{1:M}^{\neq \alpha}$.

### 2.4.2 POLY-VIEW SUFFICIENT CONTRASTIVE OBJECTIVES

As in Section 2.3.3, we begin by outlining our notation for samples from the empirical distribution. Let us assume that we have the following dataset of $K$ independent $M$-tuples:

$$\mathcal{D} = \{(\mathbf{x}_{i,\alpha}, \mathbf{X}_{i,1:M}^{\neq \alpha})\} \bigcup \{(\mathbf{x}_{j,\alpha}, \mathbf{X}_{j,1:M}^{\neq \alpha})\}_{j \neq i}^K. \tag{20}$$

Following Section 2.4.1, the goal is to distinguish between conditionals $p_{\mathbf{x}_{i,\alpha} | \mathbf{X}_{i,1:M}^{\neq \alpha}}$ and $p_{\mathbf{x}_{i,\alpha} | \mathbf{X}_{j,1:M}^{\neq \gamma}}$ for any $j \neq i$ and $\gamma$, i.e. classify $\mathbf{x}_{i,\alpha}$ correctly $\forall\, i \in [K]$, giving the following procedure for finding the sufficient statistics $T^\star$ and $Q^\star$.

$$T^\star, Q^\star = \arg\max_{T,Q} \frac{p_{\mathbf{x}_{i,\alpha} | \mathbf{X}_{i,1:M}^{\neq \alpha}}}{p_{\mathbf{x}_{i,\alpha} | \mathbf{X}_{i,1:M}^{\neq \alpha}} + \sum_{j \neq i}^K \sum_{\gamma=1}^M p_{\mathbf{x}_{i,\alpha} | \mathbf{X}_{j,1:M}^{\neq \gamma}}} = \arg\max_{T,Q} \tilde{\ell}_{i,\alpha}, \tag{21}$$

leading to the the sufficient statistics contrastive loss (Equation 22),

$$\mathcal{L}_{\text{SuffStats}} = -\mathbb{E}\left[\frac{1}{K}\sum_{i=1}^{K}\frac{1}{M}\sum_{\alpha=1}^{M}\log\tilde{\ell}_{i,\alpha}\right], \quad \tilde{\ell}_{i,\alpha} = \frac{e^{T_{i,\alpha}^{\mathsf{T}}Q_{i,\tilde{\alpha}}}}{e^{T_{i,\alpha}^{\mathsf{T}}Q_{i,\tilde{\alpha}}} + \sum_{j=1}^{K}\sum_{\gamma=1}^{M}e^{T_{i,\alpha}^{\mathsf{T}}Q_{j,\tilde{\gamma}}}}, \quad (22)$$

where $\boldsymbol{x}^{\mathsf{T}}$ denotes vector transposition, $T_{i,\alpha} \equiv T(\mathbf{x}_{i,\alpha})$, and $Q_{i,\tilde{\alpha}} \equiv Q(\mathbf{X}_{i,1:M}^{\neq\alpha})$.

**Designing $Q$**   As $Q$ parameterizes the conditional (Equation 17), it is *reorderable*. Choices for $Q$ include DeepSets (Zaheer et al., 2017) and Transformers (Vaswani et al., 2017). Requiring $M = 2$ to recover SimCLR (Chen et al., 2020a) implies $Q(\mathbf{x}) = T(\mathbf{x})$, so for simplicity, we restrict ourselves to pooling operators over $T$. Finally, we want the representation space to have no special direction, which translates to orthogonal invariance of the product of $T$ and $Q$

$$[\boldsymbol{O}T(\mathbf{x}_\alpha)]^{\mathsf{T}}Q\left(\{\boldsymbol{O}T(\mathbf{x}_\beta) : \beta \neq \alpha\}\right) = T(\mathbf{x}_\alpha)^{\mathsf{T}}Q\left(\{T(\mathbf{x}_\beta) : \beta \neq \alpha\}\right), \quad (23)$$

i.e. $Q$ is equivariant $Q\left(\{\boldsymbol{O}T(\mathbf{x}_\beta) : \beta \neq \alpha\}\right) = \boldsymbol{O}Q\left(\{T(\mathbf{x}_\beta) : \beta \neq \alpha\}\right)$ which is satisfied by

$$Q(\mathbf{X}_{1:M}^{\neq\alpha}) = Q\left(\{T(\mathbf{x}_\beta) : \beta \neq \alpha\}\right) = \frac{1}{M-1}\sum_{\beta\neq\alpha}^{M}T(\mathbf{x}_\beta) \equiv \overline{T}(\mathbf{X}_{1:M}^{\neq\alpha}) \equiv \overline{T}_{\tilde{\alpha}}. \quad (24)$$

With the choice $Q = \overline{T}_{\tilde{\alpha}}$, when $M = 2$, $\mathcal{L}_{\text{SuffStats}}$ (Equation 22) recovers SimCLR (see Appendix E.2 for the detailed connection), and therefore lower bounds two-view MI. For general $M$, $\mathcal{L}_{\text{SuffStats}}$ lower bounds One-vs-Rest MI (Theorem 2.4).

**Theorem 2.4** (Sufficient Statistics lower bound One-vs-Rest MI). *For any $K$, $M \geq 2$, $B = KM$, $\alpha \in [M]$, and the choice of $Q$ in Equation 24, we have (see Appendix C.7 for the proof)*

$$\mathcal{I}\left(\mathbf{x}_\alpha; \mathbf{X}_{1:M}^{\neq\alpha}\right) \geq c(B, M) + \mathbb{E}\left[\frac{1}{K}\sum_{i=1}^{K}\log\tilde{\ell}_{i,\alpha}\right], \quad (25)$$

*where $c(B, M) = \log(B - M + 1)$, the expectation is over $K$ independent samples $\mathbf{X}_{1:K,1:M}$.*

Theorem 2.4 completes the connection between Sufficient Statistics and InfoMax (Section 2.1). We note that contrary to Average and Geometric PVC (Equations 9 and 10), the Sufficient Statistics objective for $M > 2$ (Equation 25) *cannot* be written using $F^{(2)}$ as a function basis.

## 3 EXPERIMENTS

### 3.1 SYNTHETIC 1D GAUSSIAN

Our first interests are to check our intuition and to validate how well each objective bounds the One-vs-Rest MI as described in Theorems 2.2 and 2.4. We begin with a 1D Gaussian setting, which for the generative graph (Figure 1b) corresponds to Independent and Identically Distributed (i.i.d.) samples $\mathbf{c}_i \sim N(0, \sigma_0^2)$ for $i \in [K]$, $\rho$ is identity map, and views $\mathbf{x}_{i,\alpha} \sim N(\mathbf{c}_i, \sigma^2)$ for each $\alpha \in [M]$ and $i$. One can compute One-vs-Rest MI in closed form (see Appendix E.6 for the proof):

$$\mathcal{I}(\mathbf{x}_{i,\alpha}; \mathbf{X}_{i,1:M}^{\neq\alpha}) = \frac{1}{2}\log\left[\left(1 + \frac{\sigma_0^2}{\sigma^2}\right)\left(1 - \frac{\sigma_0^2}{\sigma^2 + M\sigma_0^2}\right)\right], \quad (26)$$

which, as anticipated (Section 2.1), is an *increasing* function of $M$. Using the closed form for Gaussian differential entropy, we see:

$$\limsup_{M\to\infty}\mathcal{I}(\mathbf{x}_\alpha; \mathbf{X}_{1:M}^{\neq\alpha}) = \mathrm{H}(\mathbf{x}_\alpha) - \mathrm{H}(\mathbf{x}_\alpha|\mathbf{c}) = \mathcal{I}(\mathbf{x}_\alpha; \mathbf{c}), \quad (27)$$

i.e. One-vs-Rest MI becomes a better proxy for InfoMax as $M$ increases. Finally, we can evaluate the conditional distribution for large $M$ and see (see Appendix E.6 for the proof):

$$\lim_{M\to\infty}p_{\mathbf{x}_{i,\alpha}|\mathbf{X}_{i,1:M}^{\neq\alpha}} = p_{\mathbf{x}_{i,\alpha}|\mathbf{c}_i}, \quad (28)$$

validating our first assumption for Sufficient Statistics (Section 2.4.1).

To empirically validate our claims we train a Multi-Layer Perceptron (MLP) with the architecture (`1->32, GeLU, 32->32`) using the objectives presented in Sections 2.2, 2.3.3 and 2.4 on the synthetic Gaussian setup. We use AdamW (Loshchilov & Hutter, 2019) with learning rate $5 \times 10^{-4}$ and weight decay $5 \times 10^{-3}$, generate $K = 1024$ 1D samples in each batch, $M$ views of each sample, and train each method for 200 epochs.

We compare One-vs-Rest lower bounds of these different objectives to the true value (Equation 26). In Figure 2, we see that increasing multiplicity $M$ *decreases* the MI Gap for Geometric, Arithmetic and Sufficient, with Geometric having the lowest gap, whereas for Multi-Crop, the MI Gap *increases*, validating Theorem 2.3 and Proposition 2.1. The Multi-Crop loss expectation is also $M$-invariant, whereas its variance reduces, as was proven in Section 2.2.

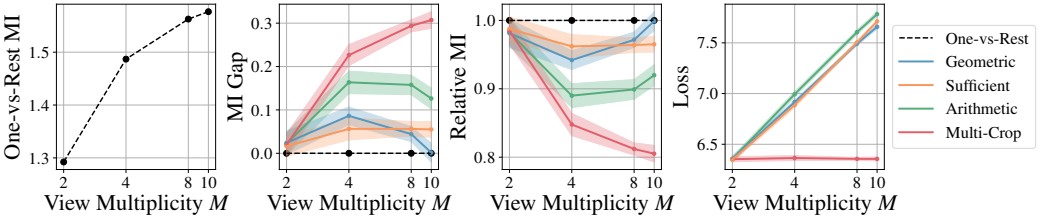

Figure 2: *Comparing MI bounds with true MI in the Gaussian setting.* Each method is trained for 200 with multiplicities $M \in \{2, 4, 8, 10\}$. Left to right: 1) True One-vs-Rest MI (Equation 26); 2) MI Gaps decrease as $M$ grows for all methods except Multi-Crop due to the $\log(K)$ factor; 3) Relative MI = True MI / Lower Bound MI; and 4) losses for each objective. Bands indicate the mean and standard deviation across 16 runs. Points indicate final model performance of corresponding hyperparameters.

## 3.2 REAL-WORLD IMAGE REPRESENTATION LEARNING

We investigate image representation learning on ImageNet1k (Russakovsky et al., 2014) following SimCLR (Chen et al., 2020a). Full experimental details are in Appendix F.1, and pseudo-code for loss calculations are in Appendix F.3.2. We consider two settings as in Fort et al. (2021):

1. *Growing Batch*, where we draw views $V = K \times M$ with multiplicity $M$ whilst preserving the number of unique samples $K$ in a batch.

2. *Fixed Batch*, where we hold the total number of views $V = K \times M$ fixed by reducing the number of unique samples $K$ as we increase the multiplicity $M$.

We investigate these scenarios at multiplicity $M = 8$ for different training epochs in Figure 3a. We observe that, given a number of training epochs *or* model updates, one should maximize view multiplicity in both *Fixed* and *Growing Batch* settings, validating the claims of Sections 2.3 and 2.4.

To understand any practical benefits, we introduce *Relative Compute*[4] (Equation 29), which is the total amount of compute used for the run compared to a SimCLR run at 128 epochs,

$$\text{Relative Compute}(M, \text{Epochs}) = \frac{M}{2} \times \frac{\text{Epochs}}{128}. \tag{29}$$

In the *Growing Batch* case, there are only minor gains with respect to the batch size 4096 SimCLR baseline when measuring relative compute. In the *Fixed Batch* case, we observe a new Pareto front in Relative Compute. *Better performance can be obtained by reducing the number of unique samples while increasing view multiplicity when using Geometric PVC or Sufficient Statistics.* Notably, a batch size 256 Geometric PVC trained for 128 epochs outperforms a batch size 4096 SimCLR trained for 1024 epochs. We also note that better performance is *not* achievable with Multi-Crop, which is compute-equivalent to SimCLR.

To further understand the role of multiplicity, we hold Epochs $= 128$ and vary multiplicity $M$ in Figure 3b. Increasing multiplicity is never harmful, with *Geometric PVC* performing the strongest overall. We note that Multi-Crop outperforms Sufficient Statistics in the *Growing Batch* setting.

---

[4]Note that there is no dependence on the number of unique samples per batch $K$, as increasing $K$ both *increases* the compute required per update step and *decreases* the number of steps per epoch.

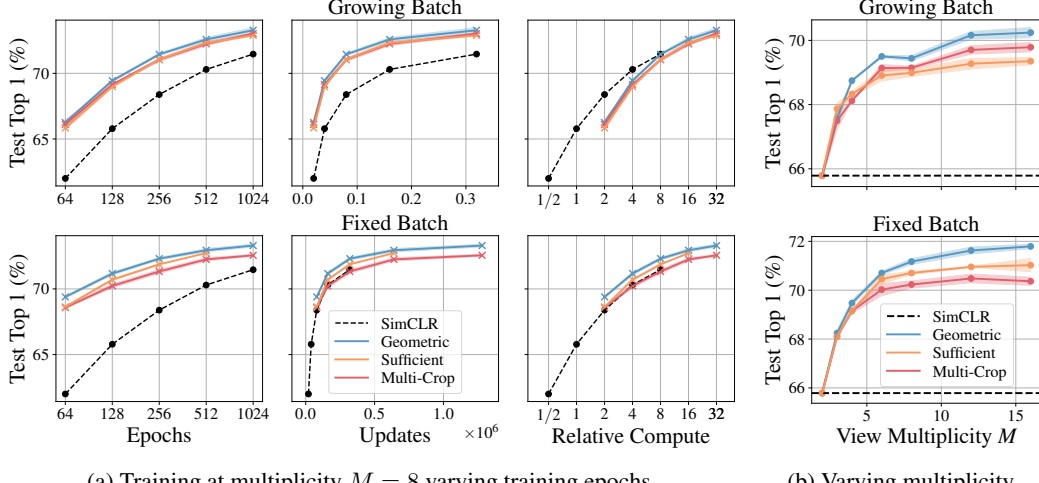

(a) Training at multiplicity $M = 8$ varying training epochs.

(b) Varying multiplicity.

Figure 3: *Contrastive ResNet 50 trained on ImageNet1k for different epochs or with different view multiplicities.* Blue, red, orange and black dashed lines represent Geometric, Multi-Crop, Sufficient Statistics, and SimCLR respectively. Bands indicate the mean and standard deviation across three runs. Points indicate *final model performance* of corresponding hyperparameters. We use $K = 4096$ for *Growing Batch* and $K = (2/M) \times 4096$ for *Fixed Batch*. (a) Each method is trained with a multiplicity $M = 8$ except the $M = 2$ SimCLR baseline. We compare models in terms of performance against training epochs (left), total updates (middle) which is affected by batch size $K$, and relative compute (right) which is defined in Equation 29. See Appendix F.3.1 for a FLOPs comparison. b) Each method is trained for 128 epochs for each multiplicity $M \in \{2, 3, 4, 6, 8, 12, 16\}$.

## 4 RELATED WORK

We present work related to view multiplicity here and additional related work in Appendix G.

**View multiplicity** Hoffer et al. (2019) showed that multiplicity improves both generalization and convergence of neural networks, helping the performance scaling. Balestriero et al. (2022b) showed that more augmentations in two-view contrastive learning helps the estimation of the MI lower-bound to have smaller variance and better convergence. Similarly, Tian et al. (2020a) studied multiple positive views in contrastive learning, however, their work enhances the loss variance by averaging over multiple *two-view* losses. While similar to the extension we present in Section 2.3.3, Tian et al. (2020a) do not consider the multiplicity effect in negatives, and the $\log(K)$ factor, resulting to just a more accurate lower-bound. Song & Ermon (2020), however, increases the $\log(K)$ factor by including positives to solve a multi-label classification problem. In the supervised setting, Fort et al. (2021) studied the effect of augmentation multiplicity in both growing and fixed batch size, showing that the signal to noise ratio increases in both cases, resulting to a better performance overall.

## 5 CONCLUSION

In self-supervised learning, the *multi* in *multi-view representation learning* typically refers to *two* views per unique sample. Given the influence of *positives*, and the *number* of *negatives* in contrastive learning, we investigated the role of the *number* of *positives*.

We showed that Multi-Crop, a popular self-supervised approach, which optimizes a combination of pair-wise tasks, reduces the variance of estimators, but *cannot* change expectations or, equivalently, bounds. To go beyond Multi-Crop, we used information theory and sufficient statistics to derive new families of representation learning methods which we call *poly-view* contrastive.

We studied the properties of these poly-view contrastive methods algorithms, and find that it is beneficial to decrease the number of unique samples whilst increasing the number of views of those samples. In particular, poly-view contrastive models trained for 128 epochs with batch size 256 outperform SimCLR trained for 1024 epochs at batch size 4096 on ImageNet1k, challenging the belief that contrastive models require large batch sizes and many training epochs.

# 6 ACKNOWLEDGEMENTS

We thank Arno Blaas, Adam Goliński, Xavier Suau, Tatiana Likhomanenko, Skyler Seto, Barry Theobald, Floris Weers, and Luca Zappella for their helpful feedback and critical discussions throughout the process of writing this paper; Okan Akalin, Hassan Babaie, Brian Gamp, Denise Hui, Mubarak Seyed Ibrahim, Li Li, Cindy Liu, Rajat Phull, Evan Samanas, Guillaume Seguin, and the wider Apple infrastructure team for assistance with developing scalable, fault tolerant code. Names are in alphabetical order by last name within group.

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

# Appendices

## A  BROADER IMPACT

This work shows different ways that *view multiplicity* can be incorporated into the design of representation learning tasks. There are a number of benefits:

1. The improved compute Pareto front shown in Section 3.2, provides a way for practitioners to achieve the desired level of model performance at *reduced* computational cost.

2. Increasing view multiplicity has a higher potential of fully capturing the aspects of a sample, as is reinforced by the limiting behavior of the synthetic setting (Section 3.1). This has the potential to learn more accurate representations for underrepresented samples.

We also note the potential undesirable consequences of our proposed methods:

1. We found that for a fixed number of updates, the best results are achieved by maximizing the multiplicity $M$. If a user is not compute limited, they may choose a high value of $M$, leading to greater energy consumption.

2. In the case one wants to maximize views that naturally occur in data as in CLIP (Radford et al., 2021), the intentional collection of additional views may be encouraged. This presents a number of challenges: 1) the collection of extensive data about a single subject increases the effort needed to collect data responsibly; 2) the collection of more than one type of data can be resource intensive; and 3) not all data collection processes are equal, and a larger number of collected views increases the chance that at least one of the views is *not* a good representation of the subject, which may negatively influence model training.

The environmental impact of each of these two points may be significant.

## B  LIMITATIONS

The work presented attempts to present a fair analysis of the different methods discussed. Despite this, we acknowledge that the work has the following limitations, which are mainly related to the real-world analysis on ImageNet1k (Section 3.2):

1. Our ImageNet1k analysis is restricted to variations of SimCLR contrastive learning method. However, there are other variations of contrastive learning, for example van den Oord et al. (2018); Chen et al. (2020b; 2021); Caron et al. (2020). There are also other types of Self-Supervised Learning (SSL) methods that train models to solve tasks involving multiple views of data, for example Grill et al. (2020); Caron et al. (2021). While we expect our results to transfer to these methods, we cannot say this conclusively.

2. Our ImageNet1k analysis is also restricted to the performance of the ResNet 50 architecture. It is possible to train SimCLR with a Vision Transformer (ViT) backbone (Chen et al., 2021; Zhai et al., 2023a), and anticipate the effect of increasing view multiplicity to be stronger in this case, as ViTs has a less strong prior on image structure, and augmentation plays a larger role in the training (Dosovitskiy et al., 2021). However, we cannot make any conclusive statements.

3. The largest number of views we consider is 16. It would be interesting to see the model behavior in for e.g. two unique samples per batch, and 2048 views per sample, or increasing the number of views beyond 16 for a larger setting. However, these settings are not practical for us to investigate, limiting the concrete statements we make for real world applications to views $M \leq 16$.

4. Although we presented some sensitivity analysis regarding augmentation policy choice in Appendix F.3, all of the augmentations we consider for ImageNet1k are variations on the SimCLR augmentation policy.

5. Our method is less applicable in the case of naturally occurring (multi-modal) data, as here $M$ is limited by the data available and *cannot* be arbitrarily increased.

6. Our empirical analysis is limited to synthetic data and the computer vision dataset ImageNet1k. While we don't anticipate significantly different conclusions for other domains, we are unable to make any conclusive empirical statements.

7. There are alternatives to One-vs-Rest Mutual Information (MI) when considering $M$ variables. We introduce an alternative partitioning in Appendix E.1, but do not investigate as it is less simple to work with.

8. In all of our experiments, hyperparameters are fixed to be those of the reference SimCLR model. In principle it is possible that a different conclusion could be drawn if a hyperparameter search was done per multiplicity configuration, and then the best performing hyperparameters for each point were compared to each other.

## C  PROOFS OF THEOREMS

### C.1  MI LOWER-BOUND WITH MULTI-CROP

**Proposition 2.1.** *For $K$ independent samples and multiplicity $M$ denoted $\mathbf{X}_{1:K,1:M}$, the Multi-Crop of any $\mathcal{L}_{Pair}$ in Equation 1 has the same MI lower bound as the corresponding $\mathcal{L}_{Pair}$*

$$\mathcal{I}(\mathbf{x}_1; \mathbf{x}_2) \geq \log(K) - \mathbb{E}\left[\mathcal{L}_{Multi\text{-}Crop}(\mathbf{X}_{1:K,1:M})\right] = \log(K) - \mathbb{E}\left[\mathcal{L}_{Pair}(\mathbf{X}_{1:K,1:2})\right], \quad (30)$$

*where the expectation is over $K$ independent samples.*

*Proof.* Note that for the pair objective $\mathcal{L}_{\text{pair}}$, we have the following lower-bound for the pair MI using $\mathcal{I}_{\text{NWJ}}$ (Hjelm et al., 2019; Nguyen et al., 2008) sample estimator:

$$\mathcal{I}(\mathbf{x}_\alpha; \mathbf{x}_\beta) \geq \log(K) - \mathbb{E}\left[\mathcal{L}_{\text{pair}}(\mathbf{X}_{1:K,\{\alpha,\beta\}})\right]. \quad (31)$$

If the views are uniformly and independently generated, i.e. $\eta_\alpha \sim \text{Uniform}(\Gamma)$, where $\Gamma$ is the set of view-generating processes, then

$$\mathcal{I}(\mathbf{x}_\alpha; \mathbf{x}_\beta) = \mathcal{I}(\mathbf{x}_\gamma; \mathbf{x}_\nu) \qquad \forall \alpha \neq \beta, \gamma \neq \nu \in [M]. \quad (32)$$

Following Equations 31 and 32, we have

$$\mathcal{I}(\mathbf{x}_1; \mathbf{x}_2) = \frac{1}{M(M-1)} \sum_{\alpha=1}^{M} \sum_{\beta \neq \alpha} \mathcal{I}(\mathbf{x}_\alpha; \mathbf{x}_\beta) \quad (33)$$

$$\geq \log(K) - \mathbb{E}\left[\frac{1}{M(M-1)} \sum_{\alpha=1}^{M} \sum_{\beta \neq \alpha} \mathcal{L}_{\text{pair}}(\mathbf{X}_{1:K,\{\alpha,\beta\}})\right] \quad (34)$$

$$= \log(K) - \mathbb{E}\left[\mathcal{L}_{\text{Multi-Crop}}(\mathbf{X}_{1:K,1:M})\right]. \quad (35)$$

Moreover, we can rewrite the Multi-Crop objective as follows in expectation:

$$\mathbb{E}\left[\mathcal{L}_{\text{Multi-Crop}}(\mathbf{X}_{1:K,1:M})\right] = \mathbb{E}\left[\frac{1}{M(M-1)} \sum_{\alpha=1}^{M} \sum_{\beta \neq \alpha} \mathcal{L}_{\text{pair}}(\mathbf{X}_{1:K,\{\alpha,\beta\}})\right] \quad (36)$$

$$= \mathbb{E}\left[\mathbb{E}_\Gamma\left[\mathcal{L}_{\text{pair}}(\mathbf{X}_{1:K,1:2})\right]\right], \quad (37)$$

where the second equality is due to the fact that all the views are uniformly and independently sampled from the set $\Gamma$. Now, getting expectation over all the randomness lead us to

$$\mathbb{E}\left[\mathcal{L}_{\text{Multi-Crop}}(\mathbf{X}_{1:K,1:M})\right] = \mathbb{E}\left[\mathbb{E}_\Gamma\left[\mathcal{L}_{\text{pair}}(\mathbf{X}_{1:K,1:2})\right]\right] = \mathbb{E}\left[\mathcal{L}_{\text{pair}}(\mathbf{X}_{1:K,1:2})\right]. \quad (38)$$

This completes the proof. $\qquad \square$

### C.2  LOWER VARIANCE OF MULTI-CROP MI BOUND

**Proposition 2.2.** *For $K$ independent samples and multiplicity $M$, $M \geq 3$, denoted $\mathbf{X}_{1:K,1:M}$, the Multi-Crop of any $\mathcal{L}_{Pair}$ in Equation 1 has a lower sample variance than the corresponding $\mathcal{L}_{Pair}$*

$$\text{Var}\left[\mathcal{L}_{Multi\text{-}Crop}(\mathbf{X}_{1:M})\right] \leq \frac{2(2M-1)}{3M(M-1)} \text{Var}\left[L_{Pair}(\mathbf{x}_1, \mathbf{x}_2)\right] < \text{Var}\left[L_{Pair}(\mathbf{x}_1, \mathbf{x}_2)\right], \quad (39)$$

*where the variance is over $K$ independent samples.*

*Proof.* We start with computing the variance of both side of Equation 1. Note that for any two pairs of $(\mathbf{x}_\alpha, \mathbf{x}_\beta)$ and $(\mathbf{x}_\gamma, \mathbf{x}_\nu)$ such that $\{\alpha, \beta\} \cap \{\gamma, \nu\} = \emptyset$, we have

$$\text{Cov}\left[\mathcal{L}_{\text{pair}}(\mathbf{x}_\alpha, \mathbf{x}_\beta), \mathcal{L}_{\text{pair}}(\mathbf{x}_\gamma, \mathbf{x}_\nu)\right] = 0, \tag{40}$$

where Cov denotes the covariance operator. This is due to the fact that view generation processes are conditionally independent (condition on $\mathbf{x}$). Thus, for any realization of $\mathbf{x}$, the conditional covariance would be zero, which leads to the expectation of the conditional covariance, and consequently Equation 40 be zero. We can also rewrite Equation 1 as follows:

$$\mathcal{L}_{\text{Multi-Crop}}(\mathbf{X}_{1:M}) = \frac{1}{M(M-1)} \sum_{\alpha=1}^{M} \sum_{\beta \neq \alpha}^{M} \mathcal{L}_{\text{Pair}}(\mathbf{x}_\alpha, \mathbf{x}_\beta) \tag{41}$$

$$= \frac{2}{M(M-1)} \sum_{\alpha=1}^{M} \sum_{\beta > \alpha}^{M} \mathcal{L}_{\text{Pair}}(\mathbf{x}_\alpha, \mathbf{x}_\beta). \tag{42}$$

Having the pairwise loss to be symmetric, we can now compute the variance of both sides as follows:

$$\text{Var}\left[\mathcal{L}_{\text{Multi-Crop}}(\mathbf{X}_{1:M})\right] = \frac{4}{M^2(M-1)^2}\left[\sum_{\alpha=1}^{M}\sum_{\beta>\alpha}^{M}\text{Var}\left[L_{\text{Pair}}(\mathbf{x}_\alpha, \mathbf{x}_\beta)\right] + \right.$$

$$\left. 2\sum_\alpha\sum_\gamma\sum_\beta \text{Cov}\left[\mathcal{L}_{\text{Pair}}(\mathbf{x}_\alpha, \mathbf{x}_\gamma), \mathcal{L}_{\text{Pair}}(\mathbf{x}_\gamma, \mathbf{x}_\beta)\right]\right]. \tag{43}$$

One way to count the number of elements in the covariance term is to note that we can sample $\alpha, \beta,$ and $\gamma$ from $[M]$ but only one of the ordered sequence of these three is acceptable due to the ordering condition in Equation 42, which results in $\frac{M(M-1)(M-2)}{6}$ choices, where $M \geq 3$.

Another main point here is that due to the identically distributed view-generative processes,

$$\text{Var}\left[L_{\text{Pair}}(\mathbf{x}_\alpha, \mathbf{x}_\beta)\right] = \text{Var}\left[L_{\text{Pair}}(\mathbf{x}_\gamma, \mathbf{x}_\nu)\right] \qquad \forall \alpha \neq \beta, \gamma \neq \nu \in [M]. \tag{44}$$

Thus, using the variance-covariance inequality, we can write that

$$\left|\text{Cov}\left[\mathcal{L}_{\text{Pair}}(\mathbf{x}_\alpha, \mathbf{x}_\gamma), \mathcal{L}_{\text{Pair}}(\mathbf{x}_\gamma, \mathbf{x}_\beta)\right]\right| \leq \text{Var}\left[L_{\text{Pair}}(\mathbf{x}_\alpha, \mathbf{x}_\gamma)\right] = \text{Var}\left[L_{\text{Pair}}(\mathbf{x}_\gamma, \mathbf{x}_\beta)\right]. \tag{45}$$

Substituting Equations 44 and 45 in Equation 43, we have the following:

$$\text{Var}\left[\mathcal{L}_{\text{Multi-Crop}}(\mathbf{X}_{1:M})\right] \leq \frac{4}{M^2(M-1)^2}\left[\frac{M(M-1)}{2}\text{Var}\left[L_{\text{Pair}}(\mathbf{x}_1, \mathbf{x}_2)\right] + \right.$$

$$\left. 2\frac{M(M-1)(M-2)}{6}\text{Var}\left[L_{\text{Pair}}(\mathbf{x}_1, \mathbf{x}_2)\right]\right]. \tag{46}$$

Simplifying the right hand side, we get

$$\text{Var}\left[\mathcal{L}_{\text{Multi-Crop}}(\mathbf{X}_{1:M})\right] \leq \frac{2(2M-1)}{3M(M-1)}\text{Var}\left[L_{\text{Pair}}(\mathbf{x}_1, \mathbf{x}_2)\right] < \text{Var}\left[L_{\text{Pair}}(\mathbf{x}_1, \mathbf{x}_2)\right], \tag{47}$$

for any $M \geq 3$. If $M = 2$, the claim is trivial as both sides are equal. Thus, the proof is complete and Multi-Crop objective has strictly lower variance compared to the pair objective in the presence of view multiplicity. $\square$

## C.3   GENERALIZED $\mathcal{I}_{\text{NWJ}}$

**Theorem 2.1.** *For any $M \geq 2$, $\alpha \in [M]$, a set of $M$ random variables $\mathbf{X}_{1:M}$, and for any positive function $F^{(M)} : \mathcal{X} \times \mathcal{X}^{M-1} \mapsto \mathbb{R}^+$*

$$\mathcal{I}(\mathbf{x}_\alpha; \mathbf{X}_{1:M}^{\neq\alpha}) \geq \mathbb{E}_{p_{\mathbf{X}_{1:M}}}\left[F^{(M)}(\boldsymbol{x}_\alpha, \boldsymbol{X}_{1:M}^{\neq\alpha})\right] - \mathbb{E}_{p_{\mathbf{x}_\alpha} p_{\mathbf{X}_{1:M}^{\neq\alpha}}}\left[e^{F^{(M)}(\boldsymbol{x}_\alpha, \boldsymbol{X}_{1:M}^{\neq\alpha})}\right] + 1 = \mathcal{I}_{\text{GenNWJ}}. \tag{48}$$

*Proof.* We start by the definition of MI:

$$\mathcal{I}\left(\mathbf{x}_\alpha; \mathbf{X}_{1:M}^{\neq\alpha}\right) = \mathbb{E}_{p_{\mathbf{x}_{1:M}}} \left[ \log \frac{p(\boldsymbol{x}_\alpha, \boldsymbol{X}_{1:M}^{\neq\alpha})}{p(\boldsymbol{x}_\alpha)p(\boldsymbol{X}_{1:M}^{\neq\alpha})} \right] \tag{49}$$

$$= \mathbb{E}_{p_{\mathbf{x}_{1:M}}} \left[ \log \frac{p(\boldsymbol{x}_\alpha, \boldsymbol{X}_{1:M}^{\neq\alpha})e^{F^{(M)}(\boldsymbol{x}_\alpha, \boldsymbol{X}_{1:M}^{\neq\alpha})}}{p(\boldsymbol{x}_\alpha)p(\boldsymbol{X}_{1:M}^{\neq\alpha})e^{F^{(M)}(\boldsymbol{x}_\alpha, \boldsymbol{X}_{1:M}^{\neq\alpha})}} \right] \tag{50}$$

$$= \mathbb{E}_{p_{\mathbf{x}_{1:M}}} \left[ F^{(M)}(\boldsymbol{x}_\alpha, \boldsymbol{X}_{1:M}^{\neq\alpha}) \right] - \mathbb{E}_{p_{\mathbf{x}_{1:M}}} \left[ \log \frac{p(\boldsymbol{x}_\alpha)p(\boldsymbol{X}_{1:M}^{\neq\alpha})e^{F^{(M)}(\boldsymbol{x}_\alpha, \boldsymbol{X}_{1:M}^{\neq\alpha})}}{p(\boldsymbol{x}_\alpha, \boldsymbol{X}_{1:M}^{\neq\alpha})} \right] \tag{51}$$

Now, we note that the argument of the second term of right hand side in Equation 51 is always positive. For any $z \geq 0$, we have that $\log(z) \leq z - 1$. Thus, we have:

$$\mathcal{I}\left(\mathbf{x}_\alpha; \mathbf{X}_{1:M}^{\neq\alpha}\right) = \mathbb{E}_{p_{\mathbf{x}_{1:M}}} \left[ F^{(M)}(\boldsymbol{x}_\alpha, \boldsymbol{X}_{1:M}^{\neq\alpha}) \right] - \mathbb{E}_{p_{\mathbf{x}_{1:M}}} \left[ \log \frac{p(\boldsymbol{x}_\alpha)p(\boldsymbol{X}_{1:M}^{\neq\alpha})e^{F^{(M)}(\boldsymbol{x}_\alpha, \boldsymbol{X}_{1:M}^{\neq\alpha})}}{p(\boldsymbol{x}_\alpha, \boldsymbol{X}_{1:M}^{\neq\alpha})} \right] \tag{52}$$

$$\geq \mathbb{E}_{p_{\mathbf{x}_{1:M}}} \left[ F^{(M)}(\boldsymbol{x}_\alpha, \boldsymbol{X}_{1:M}^{\neq\alpha}) \right] - \mathbb{E}_{p_{\mathbf{x}_{1:M}}} \left[ \frac{p(\boldsymbol{x}_\alpha)p(\boldsymbol{X}_{1:M}^{\neq\alpha})}{p(\boldsymbol{x}_\alpha, \boldsymbol{X}_{1:M}^{\neq\alpha})} e^{F^{(M)}(\boldsymbol{x}_\alpha, \boldsymbol{X}_{1:M}^{\neq\alpha})} \right] + 1. \tag{53}$$

Now, we can use the change of measure for the second term on the right hand side and the proof is complete:

$$\mathcal{I}\left(\mathbf{x}_\alpha; \mathbf{X}_{1:M}^{\neq\alpha}\right) \geq \mathbb{E}_{p_{\mathbf{x}_{1:M}}} \left[ F^{(M)}(\boldsymbol{x}_\alpha, \boldsymbol{X}_{1:M}^{\neq\alpha}) \right] - \mathbb{E}_{p_{\mathbf{x}_{1:M}}} \left[ \frac{p(\boldsymbol{x}_\alpha)p(\boldsymbol{X}_{1:M}^{\neq\alpha})e^{F^{(M)}(\boldsymbol{x}_\alpha, \boldsymbol{X}_{1:M}^{\neq\alpha})}}{p(\boldsymbol{x}_\alpha, \boldsymbol{X}_{1:M}^{\neq\alpha})} \right] + 1 \tag{54}$$

$$= \mathbb{E}_{p_{\mathbf{x}_{1:M}}} \left[ F^{(M)}\left(\boldsymbol{x}_\alpha, \boldsymbol{X}_{1:M}^{\neq\alpha}\right) \right] - \mathbb{E}_{p_{\mathbf{x}_\alpha} p_{\mathbf{X}_{1:M}^{\neq\alpha}}} \left[ e^{F^{(M)}\left(\boldsymbol{x}_\alpha, \boldsymbol{X}_{1:M}^{\neq\alpha}\right)} \right] + 1 \tag{55}$$

$$= \mathcal{I}_{\text{GenNWJ}}. \tag{56}$$

$\square$

## C.4 VALIDITY PROPERTY

**Theorem C.1.** *Both aggregation functions introduced by Equation 9 and Equation 10 satisfy the Validity property, i.e. Equation 8.*

*Proof.* Let us define $\boldsymbol{z}_\beta = \exp(F^{(2)}(\boldsymbol{x}_\alpha, \boldsymbol{x}_\beta))$ for a given $\boldsymbol{x}_\alpha$ and $\beta \neq \alpha$. Thus, we can rewrite Equations 9 and 10 as follows:

$$\text{Arithmetic:} \quad F^{(M)}\left(\boldsymbol{x}_\alpha, \boldsymbol{X}_{1:M}^{\neq\alpha}\right) = \log\left( \frac{1}{M-1} \sum_{\beta \neq \alpha} \boldsymbol{z}_\beta \right), \tag{57}$$

$$\text{Geometric:} \quad F^{(M)}\left(\boldsymbol{x}_\alpha, \boldsymbol{X}_{1:M}^{\neq\alpha}\right) = \frac{1}{M-1} \sum_{\beta \neq \alpha} \log(\boldsymbol{z}_\beta). \tag{58}$$

Following the definition of the aggregation function, and denoting $c_\alpha = \frac{p_{\mathbf{x}_\alpha} p_{\mathbf{X}_{1:M}^{\neq\alpha}}}{p_{\mathbf{x}_{1:M}}}$, we can rewrite the aggregation functions as following:

$$\text{Arithmetic:} \quad g_\alpha^{(M)} = c_\alpha \exp\left( \log\left( \frac{1}{M-1} \sum_{\beta \neq \alpha} \boldsymbol{z}_\beta \right) \right) = \frac{1}{M-1} \sum_{\beta \neq \alpha} c_\alpha \boldsymbol{z}_\beta, \tag{59}$$

$$\text{Geometric:} \quad g_\alpha^{(M)} = c_\alpha \exp\left( \frac{1}{M-1} \sum_{\beta \neq \alpha} \log(\boldsymbol{z}_\beta) \right) = \left( \prod_{\beta \neq \alpha} c_\alpha \boldsymbol{z}_\beta \right)^{\frac{1}{M-1}}. \tag{60}$$

Now, to prove the Validity for these two aggregation functions, it is enough to show the following:

$$\text{Arithmetic:} \qquad \mathcal{G}_{\text{MI}}\left(\frac{1}{M-1}\sum_{\beta\neq\alpha}c_{\alpha}\boldsymbol{z}_{\beta}\right) \leq \frac{1}{M-1}\sum_{\beta\neq\alpha}\mathcal{G}_{\text{MI}}\left(c_{\alpha}\boldsymbol{z}_{\beta}\right), \qquad (61)$$

$$\text{Geometric:} \qquad \mathcal{G}_{\text{MI}}\left(\left(\prod_{\beta\neq\alpha}c_{\alpha}\boldsymbol{z}_{\beta}\right)^{\frac{1}{M-1}}\right) \leq \frac{1}{M-1}\sum_{\beta\neq\alpha}\mathcal{G}_{\text{MI}}\left(c_{\alpha}\boldsymbol{z}_{\beta}\right). \qquad (62)$$

We start by proving Equation 61. Following the definition of MI Gap in Equation 7, we note that the MI Gap is a convex function since $g_{\alpha}-\log(g_{\alpha})-1$ is convex. Now, using the Jensen's inequality, we have:

$$\mathcal{G}_{\text{MI}}\left(\mathbb{E}_{\mathbf{z}}\left[c_{\alpha}\boldsymbol{z}\right]\right) \leq \mathbb{E}_{\mathbf{z}}\left[\mathcal{G}_{\text{MI}}\left(c_{\alpha}\boldsymbol{z}\right)\right], \qquad (63)$$

which is another expression of Equation 61 and completes the proof for Arithmetic mean. For the Geometric mean, by expanding on the definition of MI Gap in Equation 62, and removing the constant 1 from both sides, we get the following inequality:

$$\mathbb{E}\left[\left(\prod_{\beta\neq\alpha}c_{\alpha}\boldsymbol{z}_{\beta}\right)^{\frac{1}{M-1}} - \log\left(\prod_{\beta\neq\alpha}c_{\alpha}\boldsymbol{z}_{\beta}\right)^{\frac{1}{M-1}}\right] \leq \frac{1}{M-1}\sum_{\beta\neq\alpha}\mathbb{E}\left[c_{\alpha}\boldsymbol{z}_{\beta}-\log\left(c_{\alpha}\boldsymbol{z}_{\beta}\right)\right] \qquad (64)$$

$$= \mathbb{E}\left[\frac{1}{M-1}\sum_{\beta\neq\alpha}\left(c_{\alpha}\boldsymbol{z}_{\beta}-\log\left(c_{\alpha}\boldsymbol{z}_{\beta}\right)\right)\right]. \qquad (65)$$

So, proving Equation 62 is equivalent to prove the following:

$$\mathbb{E}\left[\left(\prod_{\beta\neq\alpha}c_{\alpha}\boldsymbol{z}_{\beta}\right)^{\frac{1}{M-1}} - \log\left(\prod_{\beta\neq\alpha}c_{\alpha}\boldsymbol{z}_{\beta}\right)^{\frac{1}{M-1}} - \frac{1}{M-1}\sum_{\beta\neq\alpha}\left(c_{\alpha}\boldsymbol{z}_{\beta}-\log\left(c_{\alpha}\boldsymbol{z}_{\beta}\right)\right)\right] \leq 0. \qquad (66)$$

We show for any realization of $\mathbf{z}_{\beta}$, the inequality is true, then the same applies to the expectation and the proof is complete. Note that $\log\left(\prod_{\beta\neq\alpha}c_{\alpha}\boldsymbol{z}_{\beta}\right)^{\frac{1}{M-1}} = \frac{1}{M-1}\sum_{\beta\neq\alpha}\log\left(c_{\alpha}\boldsymbol{z}_{\beta}\right)$, moreover, using arithmetic-geometric inequality for any non-negative values of $\mathbf{z}_{\beta}$ and $c_{\alpha}$, we have:

$$\left(\prod_{\beta\neq\alpha}c_{\alpha}\boldsymbol{z}_{\beta}\right)^{\frac{1}{M-1}} \leq \frac{1}{M-1}\sum_{\beta\neq\alpha}c_{\alpha}\boldsymbol{z}_{\beta}, \qquad (67)$$

which proves Equation 66, and completes the proof. $\qquad\square$

## C.5 Arithmetic and Geometric PVC

**Theorem 2.2.** *For any $K$, $M \geq 2$, $B = KM$, $\alpha \in [M]$, any scalar function $f : \mathcal{C} \times \mathcal{C} \mapsto \mathbb{R}$, and map $h : \mathcal{X} \mapsto \mathcal{C}$, we have*

$$\text{Arithmetic PVC:} \qquad \mathcal{I}\left(\mathbf{x}_{\alpha}; \mathbf{X}_{1:M}^{\neq\alpha}\right) \geq c(B,M) + \mathbb{E}\left[\frac{1}{K}\sum_{i=1}^{K}\log\frac{1}{M-1}\sum_{\beta\neq\alpha}\ell_{i,\alpha,\beta}\right], \qquad (68)$$

$$\text{Geometric PVC:} \qquad \mathcal{I}\left(\mathbf{x}_{\alpha}; \mathbf{X}_{1:M}^{\neq\alpha}\right) \geq c(B,M) + \mathbb{E}\left[\frac{1}{K}\sum_{i=1}^{K}\frac{1}{M-1}\sum_{\beta\neq\alpha}\log\ell_{i,\alpha,\beta}\right], \qquad (69)$$

*where $c(B,M) = \log(B-M+1)$, the expectation is over $K$ independent samples $\mathbf{X}_{1:K,1:M}$, and*

$$\ell_{i,\alpha,\beta}\left(\mathbf{X}_{1:K,1:M}\right) = \frac{e^{f(\widetilde{\mathbf{x}}_{i,\alpha},\widetilde{\mathbf{x}}_{i,\beta})}}{e^{f(\widetilde{\mathbf{x}}_{i,\alpha},\widetilde{\mathbf{x}}_{i,\beta})} + \sum_{j\neq i}\sum_{\gamma=1}^{M}e^{f(\widetilde{\mathbf{x}}_{j,\gamma},\widetilde{\mathbf{x}}_{i,\beta})}}, \qquad \widetilde{\mathbf{x}}_{i,\alpha} = h(\mathbf{x}_{i,\alpha}). \qquad (70)$$

*We have written $\ell_{i,\alpha,\beta}$ instead of $\ell_{i,\alpha,\beta}\left(\mathbf{X}_{1:K,1:M}\right)$ where the meaning is clear.*

*Proof.* Let us sample $K$ independent sets of $\mathbf{X}_{i,1:M}$, where $i$ denotes the sample number for $i \in [K]$. By independent here, we mean $\forall i \neq j, \forall \beta, \gamma; \mathbf{X}_{i,\beta} \perp\!\!\!\perp \mathbf{X}_{j,\gamma}$. Now, let us define $\tilde{\mathbf{X}}_{i,\alpha}$ as following:

$$\tilde{\mathbf{X}}_{i,\alpha} = \{\mathbf{X}_{j,\beta}\}_{j \neq i, \beta} \bigcup \{\mathbf{x}_{i,\alpha}\}. \tag{71}$$

Since the samples are i.i.d and the views of different samples are also independent, then $\tilde{\mathbf{X}}_{i,\alpha}$ has no more information than $\mathbf{X}_{i,\alpha}$ about $\mathbf{X}_{i,1:M}^{\neq\alpha}$. Thus,

$$\mathcal{I}(\mathbf{x}_{i,\alpha}; \mathbf{X}_{i,1:M}^{\neq\alpha}) = \mathcal{I}(\mathbf{X}_{i,\alpha}; \mathbf{X}_{i,1:M}^{\neq\alpha}) = \mathcal{I}(\tilde{\mathbf{X}}_{i,\alpha}; \mathbf{X}_{i,1:M}^{\neq\alpha}). \tag{72}$$

Moreover, since the samples are identically distributed, we have:

$$\mathcal{I}(\mathbf{x}_{\alpha}; \mathbf{X}_{1:M}^{\neq\alpha}) = \frac{1}{K}\sum_{i=1}^{K}\mathcal{I}(\mathbf{x}_{i,\alpha}; \mathbf{X}_{i,1:M}^{\neq\alpha}) = \frac{1}{K}\sum_{i=1}^{K}\mathcal{I}(\tilde{\mathbf{X}}_{i,\alpha}; \mathbf{X}_{i,1:M}^{\neq\alpha}) \tag{73}$$

Now, following the proof of Theorem 2.1, we need to define $F^{(M)}\left(\tilde{\mathbf{X}}_{i,\alpha}, \mathbf{X}_{i,1:M}^{\neq\alpha}\right)$. Following the Arithmetic and Geometric mean in Equations 9 and 10, we only need to define $F^{(2)}\left(\tilde{\mathbf{X}}_{i,\alpha}, \boldsymbol{x}_{i,\beta}\right)$ for $\beta \neq \alpha$ as the basis. Defining $\ell_{i,\alpha,\beta}\left(\mathbf{X}_{1:K,1:M}\right)$ as follows:

$$\ell_{i,\alpha,\beta}\left(\mathbf{X}_{1:K,1:M}\right) = \frac{e^{f(\tilde{\mathbf{x}}_{i,\alpha}, \tilde{\mathbf{x}}_{i,\beta})}}{e^{f(\tilde{\mathbf{x}}_{i,\alpha}, \tilde{\mathbf{x}}_{i,\beta})} + \sum_{j \neq i}\sum_{\gamma=1}^{M} e^{f(\tilde{\mathbf{x}}_{j,\gamma}, \tilde{\mathbf{x}}_{i,\beta})}}, \qquad \tilde{\mathbf{x}}_{i,\alpha} = h(\mathbf{x}_{i,\alpha}), \tag{74}$$

we can now define $F^{(2)}$ for both Arithmetic and Geometric as:

$$F^{(2)}\left(\tilde{\mathbf{X}}_{i,\alpha}, \boldsymbol{x}_{i,\beta}\right) = \log\left((B - M + 1)\,\ell_{i,\alpha,\beta}\left(\mathbf{X}_{1:K,1:M}\right)\right) = c(B, M) + \log \ell_{i,\alpha,\beta} \tag{75}$$

Now, substituting $F^{(M)}\left(\tilde{\mathbf{X}}_{i,\alpha}, \mathbf{X}_{i,1:M}^{\neq\alpha}\right)$ (denoted by $F^{(M)}$ for simplicity) in Theorem 2.1, we have the following:

**Arithmetic mean:**

$$\mathcal{I}(\mathbf{x}_{\alpha}; \mathbf{X}_{1:M}^{\neq\alpha}) = \frac{1}{K}\sum_{i=1}^{K}\mathcal{I}(\tilde{\mathbf{X}}_{i,\alpha}; \mathbf{X}_{i,1:M}^{\neq\alpha}) \tag{76}$$

$$\geq \mathbb{E}_{p_{\mathbf{x}_{1:K,1:M}}}\left[\frac{1}{K}\sum_{i=1}^{K}F^{(M)}\right] - \mathbb{E}_{\Pi_{j\neq i}p_{\tilde{\mathbf{x}}_j}p_{\mathbf{x}_{i,\alpha}}p_{\mathbf{x}_{i,1:M}^{\neq\alpha}}}\left[\frac{1}{K}\sum_{i=1}^{K}e^{F^{(M)}}\right] + 1 \tag{77}$$

$$= \mathbb{E}_{p_{\mathbf{x}_{1:K,1:M}}}\left[\frac{1}{K}\sum_{i=1}^{K}\log\frac{B - M + 1}{M - 1}\sum_{\beta \neq \alpha}\ell_{i,\alpha,\beta}\right]$$

$$- \mathbb{E}_{\Pi_{j\neq i}p_{\tilde{\mathbf{x}}_j}p_{\mathbf{x}_{i,\alpha}}p_{\mathbf{x}_{i,1:M}^{\neq\alpha}}}\left[\frac{1}{K}\sum_{i=1}^{K}\frac{B - M + 1}{M - 1}\sum_{\beta \neq \alpha}\ell_{i,\alpha,\beta}\right] + 1. \tag{78}$$

Noting that the expectation in Equation 78 is taking over variables independently, and noting that the samples are identically distributed, and different views are generated independently, we can replace $\mathbf{x}_{i,\beta}$ by a fixed $i$, e.g. without loss of generality, $i = 1$. Now, we can easily see that this term becomes equal to one. Thus,

$$\mathcal{I}(\mathbf{x}_{\alpha}; \mathbf{X}_{1:M}^{\neq\alpha}) \geq \mathbb{E}_{p_{\mathbf{x}_{1:K,1:M}}}\left[\frac{1}{K}\sum_{i=1}^{K}\log\frac{B - M + 1}{M - 1}\sum_{\beta \neq \alpha}\ell_{i,\alpha,\beta}\right] \tag{79}$$

$$= c(B, M) + \mathbb{E}\left[\frac{1}{K}\sum_{i=1}^{K}\log\frac{1}{M - 1}\sum_{\beta \neq \alpha}\ell_{i,\alpha,\beta}\right], \tag{80}$$

which is the claim of the theorem, and the proof is complete for Arithmetic mean.

**Geometric mean:**

$$\mathcal{I}(\mathbf{x}_\alpha; \mathbf{X}_{1:M}^{\neq\alpha}) = \frac{1}{K} \sum_{i=1}^{K} \mathcal{I}(\tilde{\mathbf{X}}_{i,\alpha}; \mathbf{X}_{i,1:M}^{\neq\alpha}) \tag{81}$$

$$\geq \mathbb{E}_{p_{\mathbf{x}_{1:K,1:M}}} \left[ \frac{1}{K} \sum_{i=1}^{K} F^{(M)} \right] - \mathbb{E}_{\Pi_{j\neq i} p_{\tilde{\mathbf{X}}_j} p_{\mathbf{x}_{i,\alpha}} p_{\mathbf{X}_{i,1:M}^{\neq\alpha}}} \left[ \frac{1}{K} \sum_{i=1}^{K} e^{F^{(M)}} \right] + 1 \tag{82}$$

$$= \mathbb{E}_{p_{\mathbf{x}_{1:K,1:M}}} \left[ c(B,M) + \frac{1}{K} \sum_{i=1}^{K} \frac{1}{M-1} \sum_{\beta\neq\alpha} \log \ell_{i,\alpha,\beta} \right] + 1$$

$$- \mathbb{E}_{\Pi_{j\neq i} p_{\tilde{\mathbf{X}}_j} p_{\mathbf{x}_{i,\alpha}} p_{\mathbf{X}_{i,1:M}^{\neq\alpha}}} \left[ \frac{1}{K} \sum_{i=1}^{K} \exp\left(c(B,M) + \frac{1}{M-1} \sum_{\beta\neq\alpha} \log \ell_{i,\alpha,\beta}\right) \right]. \tag{83}$$

Since $\exp(z)$ is a convex function, we can use the Jensen's inequality for Equation 83:

$$\mathbb{E}_{\Pi_{j\neq i} p_{\tilde{\mathbf{X}}_j} p_{\mathbf{x}_{i,\alpha}} p_{\mathbf{X}_{i,1:M}^{\neq\alpha}}} \left[ \frac{1}{K} \sum_{i=1}^{K} \exp\left(c(B,M) + \frac{1}{M-1} \sum_{\beta\neq\alpha} \log \ell_{i,\alpha,\beta}\right) \right] \tag{84}$$

$$\leq \mathbb{E}_{\Pi_{j\neq i} p_{\tilde{\mathbf{X}}_j} p_{\mathbf{x}_{i,\alpha}} p_{\mathbf{X}_{i,1:M}^{\neq\alpha}}} \left[ \frac{1}{K} \sum_{i=1}^{K} \frac{1}{M-1} \sum_{\beta\neq\alpha} (B-M+1)\ell_{i,\alpha,\beta} \right] \tag{85}$$

$$= 1.$$

Where the last equality is resulted with the same reasoning behind Equation 78. Thus, we have:

$$\mathcal{I}(\mathbf{x}_\alpha; \mathbf{X}_{1:M}^{\neq\alpha}) \geq \mathbb{E}_{p_{\mathbf{x}_{1:K,1:M}}} \left[ c(B,M) + \frac{1}{K} \sum_{i=1}^{K} \frac{1}{M-1} \sum_{\beta\neq\alpha} \log \ell_{i,\alpha,\beta} \right] + 1 - 1 \tag{86}$$

$$= c(B,M) + \mathbb{E} \left[ \frac{1}{K} \sum_{i=1}^{K} \frac{1}{M-1} \sum_{\beta\neq\alpha} \log \ell_{i,\alpha,\beta} \right], \tag{87}$$

and the proof is complete. $\qquad\square$

## C.6 BEHAVIOR OF MI GAP

To investigate the behavior of MI Gap and to provide the proof of Theorem 2.3, we first provide the following lemma, which is resulted only by the definition of expectation in probability theory:

**Lemma C.2.** *Let $I \subset \{1, \ldots, k\}$ with $|I| = m$, $m \leq k$, be a uniformly distributed subset of distinct indices from $\{1, \ldots, k\}$. Then, the following holds for any sequence of numbers $a_1, \ldots, a_k$.*

$$\mathbb{E}_{I=\{i_1,\ldots,i_m\}} \left[ \frac{a_{i_1} + \ldots + a_{i_m}}{m} \right] = \frac{a_1 + \ldots + a_k}{k} \tag{88}$$

Now, for Theorem 2.3, we have the following:

**Theorem 2.3.** *For fixed $\alpha$, the MI Gap of Arithmetic and Geometric PVC are monotonically non-increasing with $M$:*

$$\mathcal{G}_{\text{MI}}(\mathbf{X}_{1:M_2}; g_\alpha^{(M_2)}) \leq \mathcal{G}_{\text{MI}}(\mathbf{X}_{1:M_1}; g_\alpha^{(M_1)}) \quad \forall M_1 \leq M_2. \tag{89}$$

*Proof.* Let us use the new form of aggregation functions' definition with $z_\beta$ in Equations 59 and 60. For $M_1 \leq M_2$, and for Arithmetic mean, i.e. $g_\alpha^{(M)} = \frac{1}{M-1} \sum_{\beta \neq \alpha} c_\alpha z_\beta$, we have:

$$\mathcal{G}_{\mathrm{MI}}(\mathbf{X}_{1:M_2}; g_\alpha^{(M_2)}) = \mathbb{E}_{p_{\mathbf{x}_{1:M}}} \left[ \frac{1}{M_2 - 1} \sum_{\beta \neq \alpha} c_\alpha z_\beta \right]$$

$$- \mathbb{E}_{p_{\mathbf{x}_{1:M_2}}} \left[ \log \left( \frac{1}{M_2 - 1} \sum_{\beta \neq \alpha} c_\alpha z_\beta \right) \right] - 1 \qquad (90)$$

$$= \mathbb{E}_{p_{\mathbf{x}_{1:M_2}}} \left[ \mathbb{E}_{I=\{\gamma_1,...,\gamma_{M_1-1}\}} \left[ \frac{1}{M_1 - 1} \sum_{j=1}^{M_1-1} c_\alpha z_{\gamma_j} \right] \right]$$

$$- \mathbb{E}_{p_{\mathbf{x}_{1:M_2}}} \left[ \log \left( \mathbb{E}_{I=\{\gamma_1,...,\gamma_{M_1-1}\}} \left[ \frac{1}{M_1 - 1} \sum_{j=1}^{M_1-1} c_\alpha z_{\gamma_j} \right] \right) \right] - 1 \quad (91)$$

$$\leq \mathbb{E}_{p_{\mathbf{x}_{1:M_1}}} \left[ \frac{1}{M_1 - 1} \sum_{\beta \neq \alpha} c_\alpha z_\beta \right]$$

$$- \mathbb{E}_{p_{\mathbf{x}_{1:M_1}}} \left[ \mathbb{E}_{I=\{\gamma_1,...,\gamma_{M_1-1}\}} \left[ \log \left( \frac{1}{M_1 - 1} \sum_{j=1}^{M_1-1} c_\alpha z_{\gamma_j} \right) \right] \right] - 1 \quad (92)$$

$$= \mathbb{E}_{p_{\mathbf{x}_{1:M_1}}} \left[ \frac{1}{M_1 - 1} \sum_{\beta \neq \alpha} c_\alpha z_\beta \right]$$

$$- \mathbb{E}_{p_{\mathbf{x}_{1:M_1}}} \left[ \log \left( \frac{1}{M_1 - 1} \sum_{\beta \neq \alpha} c_\alpha z_\beta \right) \right] - 1 \qquad (93)$$

$$= \mathcal{G}_{\mathrm{MI}}(\mathbf{X}_{1:M_1}; g_\alpha^{(M_1)}), \qquad (94)$$

where the first equality is due to the Lemma C.2, and the inequality is resulted from Jensen's inequality. Therefore, for Arithmetic mean, the MI Gap is decreasing with respect to $M$.

For the Geometric mean, and following the definition of MI Gap, we have:

$$\mathcal{G}_{\mathrm{MI}}(\mathbf{X}_{1:M_2}; g_\alpha^{(M_2)}) = \mathbb{E}_{p_{\mathbf{x}_{1:M_2}}} \left[ \left( \prod_{\beta \neq \alpha} c_\alpha z_\beta \right)^{\frac{1}{M_2 - 1}} \right]$$

$$- \mathbb{E}_{p_{\mathbf{x}_{1:M_2}}} \left[ \frac{1}{M_2 - 1} \sum_{\beta \neq \alpha} \log (c_\alpha z_\beta) \right] - 1 \qquad (95)$$

$$= \mathbb{E}_{p_{\mathbf{x}_{1:M_2}}} \left[ \left( \prod_{\beta \neq \alpha} c_\alpha z_\beta \right)^{\frac{1}{M_2 - 1}} \right]$$

$$- \mathbb{E}_{p_{\mathbf{x}_{1:M_1}}} \left[ \frac{1}{M_1 - 1} \sum_{\beta \neq \alpha} \log (c_\alpha z_\beta) \right] - 1, \qquad (96)$$

where the equality is followed by Lemma C.2, similarly to the corresponding proof for the Arithmetic mean. Now, mainly focusing on the first term of the MI Gap, we have:

$$\mathcal{G}_{\mathrm{MI}}(\mathbf{X}_{1:M_2}; g_\alpha^{(M_2)}) = \mathbb{E}_{p_{\mathbf{x}_{1:M_2}}} \left[ \exp \log \left( \prod_{\beta \neq \alpha}^{M_2 - 1} c_\alpha \mathbf{z}_\beta \right)^{\frac{1}{M_2 - 1}} \right]$$

$$- \mathbb{E}_{p_{\mathbf{x}_{1:M_1}}} \left[ \frac{1}{M_1 - 1} \sum_{\beta \neq \alpha} \log (c_\alpha \mathbf{z}_\beta) \right] - 1 \tag{97}$$

$$= \mathbb{E}_{p_{\mathbf{x}_{1:M_2}}} \left[ \exp \frac{1}{M_2 - 1} \sum_{\beta \neq \alpha}^{M_2 - 1} \log c_\alpha \mathbf{z}_\beta \right]$$

$$- \mathbb{E}_{p_{\mathbf{x}_{1:M_1}}} \left[ \frac{1}{M_1 - 1} \sum_{\beta \neq \alpha} \log (c_\alpha \mathbf{z}_\beta) \right] - 1 \tag{98}$$

$$= \mathbb{E}_{p_{\mathbf{x}_{1:M_2}}} \left[ \exp \left( \mathbb{E}_{I = \{\gamma_1, \dots, \gamma_{M_1-1}\}} \left[ \frac{1}{M_1 - 1} \sum_{j=1}^{M_1 - 1} \log c_\alpha \mathbf{z}_{\gamma_j} \right] \right) \right]$$

$$- \mathbb{E}_{p_{\mathbf{x}_{1:M_1}}} \left[ \frac{1}{M_1 - 1} \sum_{\beta \neq \alpha} \log (c_\alpha \mathbf{z}_\beta) \right] - 1 \tag{99}$$

$$\leq \mathbb{E}_{p_{\mathbf{x}_{1:M_2}}} \left[ \mathbb{E}_{I = \{\gamma_1, \dots, \gamma_{M_1-1}\}} \left[ \exp \left( \frac{1}{M_1 - 1} \sum_{j=1}^{M_1 - 1} \log c_\alpha \mathbf{z}_{\gamma_j} \right) \right] \right]$$

$$- \mathbb{E}_{p_{\mathbf{x}_{1:M_1}}} \left[ \frac{1}{M_1 - 1} \sum_{\beta \neq \alpha} \log (c_\alpha \mathbf{z}_\beta) \right] - 1 \tag{100}$$

$$= \mathbb{E}_{p_{\mathbf{x}_{1:M_1}}} \left[ \left( \prod_{\beta \neq \alpha} c_\alpha \mathbf{z}_\beta \right)^{\frac{1}{M_1 - 1}} \right]$$

$$- \mathbb{E}_{p_{\mathbf{x}_{1:M_1}}} \left[ \frac{1}{M_1 - 1} \sum_{\beta \neq \alpha} \log (c_\alpha \mathbf{z}_\beta) \right] - 1 \tag{101}$$

$$= \mathcal{G}_{\mathrm{MI}}(\mathbf{X}_{1:M_1}; g_\alpha^{(M_1)}). \tag{102}$$

Here, Equation 100 is resulted using Lemma C.2 by replacing $a_\beta = \log c_\alpha \mathbf{z}_\beta$, and the inequality is due to the Jensen's inequality. Thus, the proof is complete. $\qquad\square$

## C.7 CONNECTION BETWEEN SUFFICIENT STATISTICS AND MI BOUNDS

**Theorem 2.4.** *For any $K$, $M \geq 2$, $B = KM$, $\alpha \in [M]$, and the choice of $Q$ in Equation 24, we have (see Appendix C.7 for the proof)*

$$\mathcal{I}\left( \mathbf{x}_\alpha; \mathbf{X}_{1:M}^{\neq \alpha} \right) \geq c(B, M) + \mathbb{E} \left[ \frac{1}{K} \sum_{i=1}^{K} \log \tilde{\ell}_{i,\alpha} \right], \tag{103}$$

*where $c(B, M) = \log(B - M + 1)$, the expectation is over $K$ independent samples $\mathbf{X}_{1:K,1:M}$.*

*Proof.* The proof consists of two parts:

1. Show that there is $F^{(M)}$ corresponding to the choice of $Q$ in Equation 24.

2. Achieving the lower-bound using the given $F^{(M)}$ for $\mathcal{I}\left(\mathbf{x}_\alpha, \mathbf{X}_{1:M}^{\neq\alpha}\right)$.

We prove both points together by studying the lower-bound for one-vs-rest MI given the aforementioned $F^{(M)}$. The proof is very similar to the proof of Theorem 2.2. We use the definition of $\tilde{\mathbf{X}}_i$ as Equation 71. We also note that since the samples are i.i.d, and the view generation is independent, we can also use Equations 72 and 73. Consequently, we only need to define the sample-based $F^{(M)}\left(\tilde{\mathbf{X}}_i, \mathbf{X}_{i,1:M}^{\neq\alpha}\right)$. Note that here, in contrast with Arithmetic and Geometric, we do not have $F^{(2)}$ as our basis for $F^{(M)}$. We define the $F^{(M)}\left(\tilde{\mathbf{X}}_i, \mathbf{X}_{i,1:M}^{\neq\alpha}\right)$ as follows:

$$F^{(M)}\left(\mathbf{X}_{1:K,1:M}; \alpha\right) = c(B, M) + \frac{1}{K}\sum_{i=1}^{K}\log \tilde{\ell}_{i,\alpha,\beta}, \tag{104}$$

which is the sample-based generalization of $F^{(M)}\left(\mathbf{x}_\alpha, \mathbf{X}_{1:M}^{\neq\alpha}\right) = T(\mathbf{x}_\alpha) \cdot \frac{\sum_{\beta\neq\alpha}^{M} T(\mathbf{x}_\beta)}{M-1}$. We also note that the introduced $F^{(M)}$ and its corresponding aggregation function, follows all the main properties, i.e. interchangeable arguments, poly-view order invariance, and expandability. Thus, the first point is correct. Now, we continue with the lower-bound. Substituting Equation 104 in Theorem 2.1, we get the following:

$$\mathcal{I}(\mathbf{x}_\alpha; \mathbf{X}_{1:M}^{\neq\alpha}) = \frac{1}{K}\sum_{i=1}^{K}\mathcal{I}(\tilde{\mathbf{X}}_i; \mathbf{X}_{i,1:M}^{\neq\alpha}) \tag{105}$$

$$\geq \mathbb{E}_{p_{\mathbf{X}_{1:K,1:M}}}\left[\frac{1}{K}\sum_{i=1}^{K}F^{(M)}\right] - \mathbb{E}_{\Pi_{j\neq i}p_{\tilde{\mathbf{X}}_j}p_{\mathbf{x}_{i,\alpha}}p_{\mathbf{X}_{i,1:M}^{\neq\alpha}}}\left[\frac{1}{K}\sum_{i=1}^{K}e^{F^{(M)}}\right] + 1 \tag{106}$$

$$= \mathbb{E}_{p_{\mathbf{X}_{1:K,1:M}}}\left[c(B, M) + \frac{1}{K}\sum_{i=1}^{K}\frac{1}{M-1}\sum_{\beta\neq\alpha}\log \tilde{\ell}_{i,\alpha,\beta}\right]$$

$$- \mathbb{E}_{\Pi_{j\neq i}p_{\tilde{\mathbf{X}}_j}p_{\mathbf{x}_{i,\alpha}}p_{\mathbf{X}_{i,1:M}^{\neq\alpha}}}\left[\frac{1}{K}\sum_{i=1}^{K}(B - M + 1)\tilde{\ell}_{i,\alpha,\beta}\right] + 1 \tag{107}$$

$$= c(B, M) + \mathbb{E}_{p_{\mathbf{X}_{1:K,1:M}}}\left[\frac{1}{K}\sum_{i=1}^{K}\frac{1}{M-1}\sum_{\beta\neq\alpha}\log \tilde{\ell}_{i,\alpha,\beta}\right], \tag{108}$$

where the last inequality is resulted with the same reasoning as having identically distributed pairs of $(\tilde{\mathbf{X}}_i, \mathbf{X}_{i,1:M}^{\neq\alpha})$ due to the sample generation process, and the fact that expectation is taken over random variables independently. Note that here, maximizing the lower-bound corresponds to maximizing $\tilde{\ell}_{i,\alpha,\beta}$, which provides the same optimization problem as Equation 21 with $Q$ in Equation 24. Thus, the proof is complete and sufficient statistics is also an MI lower-bound. $\square$

## D  NOTES ON MULTI-CROP

### D.1  DISTRIBUTION FACTORIZATION CHOICE OF MULTI-CROP

As explained in more detail in the main text, Tian et al. (2020b) and Caron et al. (2020) studied the idea of view multiplicity. While their technical approach is different, they both took a similar approach of multiplicity; getting average of pairwise (two-view) objectives as in Equation 1. Here, we show that this choice of combining objectives inherently applies a specific choice of factorization to the estimation of true conditional distribution of $p(\mathbf{x}|\mathbf{c})$ in Figure 1b using multiple views, i.e. applies an inductive bias in the choice of distribution factorization.

Following the InfoMax objective, we try to estimate $\mathcal{I}(\mathbf{x}; \mathbf{c})$ using the pairwise proxy $\mathcal{I}(\mathbf{x}_\alpha; \mathbf{x}_\beta)$. Thus, the idea of Multi-Crop can be written as following inequalities:

$$\mathcal{I}(\mathbf{x}; \mathbf{c}) \geq \frac{1}{M}\sum_{\alpha=1}^{M}\mathcal{I}(\mathbf{x}; \mathbf{x}_\alpha) \geq \frac{1}{M(M-1)}\sum_{\alpha=1}^{M}\sum_{\beta\neq\alpha}^{M}\mathcal{I}(\mathbf{x}_\alpha; \mathbf{x}_\beta). \tag{109}$$

Assuming that these two lower-bound terms are estimations for the left hand side applies distributional assumption. To see this, we start with expanding on the MI definition in each term:

$$\mathcal{I}(\mathbf{x}; \mathbf{c}) = \mathbb{E}\left[\log \frac{p(\mathbf{x}|\mathbf{c})}{p(\mathbf{x})}\right] \tag{110}$$

$$\frac{1}{M}\sum_{\alpha=1}^{M}\mathcal{I}(\mathbf{x};\mathbf{x}_\alpha) = \frac{1}{M}\sum_{\alpha=1}^{M}\mathbb{E}\left[\log\frac{p(\mathbf{x}|\mathbf{x}_\alpha)}{p(\mathbf{x})}\right] \tag{111}$$

$$\frac{1}{M(M-1)}\sum_{\alpha=1}^{M}\sum_{\beta\neq\alpha}^{M}\mathcal{I}(\mathbf{x}_\alpha;\mathbf{x}_\beta) = \frac{1}{M(M-1)}\sum_{\alpha=1}^{M}\sum_{\beta\neq\alpha}^{M}\mathbb{E}\left[\log\frac{p(\mathbf{x}_\alpha|\mathbf{x}_\beta)}{p(\mathbf{x}_\alpha)}\right]. \tag{112}$$

Now, assuming that the view-generative processes do not change the marginal distributions, i.e. for any $\alpha$, $p(\mathbf{x}) = p(\mathbf{x}_\alpha)$, and considering Equation 109, we have:

$$\mathbb{E}\left[\log\frac{p(\mathbf{x}|\mathbf{c})}{p(\mathbf{x})}\right] \geq \mathbb{E}\left[\log\left(\prod_{\alpha=1}^{M}\frac{p(\mathbf{x}|\mathbf{x}_\alpha)}{p(\mathbf{x})}\right)^{\frac{1}{M}}\right] \geq \mathbb{E}\left[\log\left(\prod_{\alpha=1}^{M}\prod_{\beta\neq\alpha}^{M}\frac{p(\mathbf{x}_\alpha|\mathbf{x}_\beta)}{p(\mathbf{x})}\right)^{\frac{1}{M(M-1)}}\right]. \tag{113}$$

Therefore, the distributional assumption or the choice of factorization is estimating $p(\mathbf{x}|\mathbf{c})$ by the following distributions:

$$p(\mathbf{x}|\mathbf{c}) \mathrel{\widehat{=}} \left(\prod_{\alpha=1}^{M}p(\mathbf{x}|\mathbf{x}_\alpha)\right)^{\frac{1}{M}}, \tag{114}$$

$$p(\mathbf{x}|\mathbf{c}) \mathrel{\widehat{=}} \left(\prod_{\alpha=1}^{M}\prod_{\beta\neq\alpha}^{M}p(\mathbf{x}_\alpha|\mathbf{x}_\beta)\right)^{\frac{1}{M(M-1)}}, \tag{115}$$

which translates to estimating the distribution using its geometric mean. Note that the symbol $\widehat{=}$ reads as "estimates", and it is not equality.

## D.2 Aggregation Function for Multi-Crop

Following the result of Proposition 2.1, Multi-Crop is an average of pairwise objectives, which means that if we know the aggregation function and $F^{(2)}$ for the pairwise objective, then we can write the aggregation function for multi-crop as following:

$$\mathcal{I}(\mathbf{x}_\alpha; \mathbf{X}_{1:M}^{\neq\alpha}) \geq \frac{1}{M-1}\sum_{\beta\neq\alpha}\mathcal{I}(\mathbf{x}_\alpha;\mathbf{x}_\beta), \tag{116}$$

$$\mathcal{I}(\mathbf{x}_\alpha;\mathbf{x}_\beta) \geq \mathbb{E}\left[F(\boldsymbol{x}_\alpha,\boldsymbol{x}_\beta)\right] - \mathbb{E}\left[\frac{p(\boldsymbol{x}_\alpha)p(\boldsymbol{x}_\beta)e^{F(\boldsymbol{x}_\alpha,\boldsymbol{x}_\beta)}}{p(\boldsymbol{x}_\alpha,\boldsymbol{x}_\beta)}\right] + 1, \tag{117}$$

where the second line is from Theorem 2.1 by setting $M = 2$. Now, by getting average over $\beta$ from both sides, we can see:

$$\frac{\sum_{\beta\neq\alpha}\mathcal{I}(\mathbf{x}_\alpha;\mathbf{x}_\beta)}{M-1} \geq \mathbb{E}_\beta\left[\mathbb{E}\left[F(\boldsymbol{x}_\alpha,\boldsymbol{x}_\beta)\right]\right] - \mathbb{E}\left[\frac{1}{M-1}\sum_{\beta\neq\alpha}\frac{p(\boldsymbol{x}_\alpha)p(\boldsymbol{x}_\beta)e^{F(\boldsymbol{x}_\alpha,\boldsymbol{x}_\beta)}}{p(\boldsymbol{x}_\alpha,\boldsymbol{x}_\beta)}\right] + 1. \tag{118}$$

Following the proof of Theorem 2.1 and the definition of aggregation function in Equation 6, we achieve the following aggregation function for Multi-Crop:

$$g_\alpha^{(M)}\left(\mathbf{X}_{1:M}^{\neq\alpha}\right) = \frac{1}{M-1}\sum_{\beta\neq\alpha}^{M}g_\alpha^{(2)}\left(\mathbf{X}_{\{\alpha,\beta\}}\right) \tag{119}$$

# E  ADDITIONAL THEORETICAL RESULTS AND DISCUSSIONS

## E.1  GENERALIZING ONE-VS-REST MI TO SETS

A generalization of the one-vs-rest MI is to consider the MI between two sets. Let us assume that the set of $[M]$ is partitioned into two sets $\mathbb{A}$ and $\mathbb{B}$, i.e. $\mathbf{X}_\mathbb{A} \cup \mathbf{X}_\mathbb{B} = \mathbf{X}_{1:M}$, and $\mathbb{A} \cap \mathbb{B} = \emptyset$, where $\emptyset$ denotes the empty set. Defining the density of $\mathbf{X}_\mathbb{A}$ and $\mathbf{X}_\mathbb{B}$ as the joint distribution of their corresponding random variables, we can define the generalized version of one-vs-rest MI:

**Definition E.1** (Two-Set MI). *For any two partition set of $\mathbb{A}$ and $\mathbb{B}$ over $[M]$, define the two-set MI as following:*

$$\mathcal{I}(\mathbf{X}_\mathbb{A}, \mathbf{X}_\mathbb{B}) = \mathcal{D}_{\mathrm{KL}}\left(p_{\mathbf{X}_{1:M}} \parallel p_{\mathbf{X}_\mathbb{A}} p_{\mathbf{X}_\mathbb{B}}\right). \tag{120}$$

We can now also generalize the $\mathcal{I}_{\mathrm{NWJ}}$ to the two-set case. The main change here is the definition of $F^{(M)}$ as it needs to be defined over two sets as inputs. We have the following:

**Theorem E.1.** *For any $M \geq 2$, and partition sets $\mathbb{A}$ and $\mathbb{B}$ over $[M]$, such that $\mathbb{A} \neq \emptyset$ and $\mathbb{B} \neq \emptyset$, and for any positive function $F^{(M)} : \mathcal{X}^{|\mathbb{A}|} \times \mathcal{X}^{|\mathbb{B}|} \mapsto \mathbb{R}^+$, we have:*

$$\mathcal{I}\left(\mathbf{X}_\mathbb{A}; \mathbf{X}_\mathbb{B}\right) \geq \mathbb{E}_{p_{\mathbf{X}_{1:M}}}\left[F^{(M)}\left(\mathbf{X}_\mathbb{A}, \mathbf{X}_\mathbb{B}\right)\right] - \mathbb{E}_{p_{\mathbf{X}_\mathbb{A}} p_{\mathbf{X}_\mathbb{B}}}\left[e^{F^{(M)}(\mathbf{X}_\mathbb{A}, \mathbf{X}_\mathbb{B})}\right] + 1. \tag{121}$$

*Proof.* The proof follows the exact proof of Theorem 2.1 by replacing $\mathbf{x}_\alpha$ and $\mathbf{X}_{1:M}^{\neq\alpha}$ by $\mathbf{X}_\mathbb{A}$ and $\mathbf{X}_\mathbb{B}$, respectively. □

Thus, as long as one can define such a function $F^{(M)}$, the other results of this paper follows.

## E.2  RECOVERING SIMCLR

**Geometric and Arithmetic PVC**  Here, we show that in case of $M = 2$ and for specific choices of function $F^{(2)}$, we can recover the existing loss objective for SimCLR, i.e. InfoNCE. Setting $M = 2$, we make the following observations:

- In the case of $M = 2$, the arithmetic and geometric aggregation functions result in the same lower-bound.
- **Recovering SimCLR** Chen et al. (2020a): Substituting $M = 2$ in Equations 12 and 13, we recover the following contrastive loss, which is equivalent to InfoNCE, i.e. SimCLR objective:

$$\mathcal{L}_{\mathrm{PVC}}^{M=2} = -\mathbb{E}\left[\frac{1}{K}\sum_{i=1}^{K} \log \frac{e^{f(\mathbf{x}_{i,1}, \mathbf{x}_{i,2})}}{e^{f(\mathbf{x}_{i,1}, \mathbf{x}_{i,2})} + \sum_{j\neq i}\sum_{\gamma=1}^{2} e^{f(\mathbf{x}_{j,\gamma}, \mathbf{x}_{i,2})}}\right] = \mathcal{L}_{\mathrm{InfoNCE}}. \tag{122}$$

Letting $f(\mathbf{x}, \mathbf{y}) = \frac{\mathbf{x} \cdot \mathbf{y}}{\|\mathbf{x}\|\|\mathbf{y}\|}$ lead us to the exact SimCLR loss.

**Sufficient Statistics**  We can also show that in the sufficient statistics loss (Equation 22), the case of $M = 2$ and the choice of $Q = \overline{T}_{\tilde{\alpha}}$ (Equation 24) recovers the SimCLR loss. To prove this, note the following observations:

- When $M = 2$, $\mathbf{X}_{1:M}^{\neq\alpha} = \mathbf{x}_{3-\alpha}$, i.e. if $\alpha = 1$, $\mathbf{X}_{1:M}^{\neq\alpha} = \mathbf{x}_2$ and if $\alpha = 2$, $\mathbf{X}_{1:M}^{\neq\alpha} = \mathbf{x}_1$.
- By Equation 24, $Q(\mathbf{X}_{1:M}^{\neq\alpha}) = \frac{1}{M-1}\sum_{\beta\neq\alpha}^{M} T(\mathbf{x}_\beta) = T(\mathbf{x}_{3-\alpha})$ when $M = 2$. Therefore, in Equation 22, we have the following:

$$\mathcal{L}_{\mathrm{SuffStats}} = -\mathbb{E}\left[\frac{1}{K}\sum_{i=1}^{K}\frac{1}{2}\sum_{\alpha=1}^{2} \log \frac{e^{T_{i,\alpha}^\intercal T_{i,3-\alpha}}}{e^{T_{i,\alpha}^\intercal T_{i,3-\alpha}} + \sum_{j=1}^{K}\sum_{\gamma=1}^{2} e^{T_{i,\alpha}^\intercal T_{j,\gamma}}}\right]$$

$$= -\mathbb{E}\left[\frac{1}{K}\sum_{i=1}^{K}\frac{1}{2}\left(\log \frac{e^{T_{i,1}^\intercal T_{i,2}}}{e^{T_{i,1}^\intercal T_{i,2}} + \sum_{j=1}^{K}\sum_{\gamma=1}^{2} e^{T_{i,1}^\intercal T_{j,\gamma}}}\right.\right. \tag{123}$$

$$\left.\left. + \log \frac{e^{T_{i,2}^\intercal T_{i,1}}}{e^{T_{i,2}^\intercal T_{i,1}} + \sum_{j=1}^{K}\sum_{\gamma=1}^{2} e^{T_{i,2}^\intercal T_{j,\gamma}}}\right)\right],$$

which is the symmetric InfoNCE. Choosing $T(\mathbf{x}) = \frac{\mathbf{x}}{\|\mathbf{x}\|}$ recovers the SimCLR objective.

### E.3 SIGLIP CONNECTION TO MI BOUND

We show that the objective introduced in Zhai et al. (2023b) is an MI bound. As of our best understanding, this is not present in the existing literature.

**SigLIP is an MI bound:** As shown in the proof of Theorem 2.2, to achieve the lower-bounds we define $F^{(M)}$ to have a Softmax-based form (see Appendix C.5 for more details). However, we could choose other forms of functions. If we replace Softmax with a Sigmoid-based form, we can recover the SigLIP loss, i.e. :

$$F^{(2)}(\boldsymbol{x}_{i,1}, \boldsymbol{x}_{j,2}) = \log \frac{1}{1 + e^{z_{i,j}(-t\boldsymbol{x}_{i,1} \cdot \boldsymbol{x}_{j,2} + b)}} \quad z_{i,j} = 1 \text{ if } (i=j) \text{ else } -1. \quad (124)$$

Following the same procedure as the proof of Theorem 2.2, and defining the $F^{(2)}$ over positives and negatives as $F^{(2)}(\mathbf{X}_{1:K,1:2}) = \frac{1}{K} \sum_{i=1}^{K} \sum_{j=1}^{K} \log \frac{1}{1 + e^{z_{i,j}(-t\mathbf{x}_{i,1} \cdot \mathbf{x}_{j,2} + b)}}$. This shows that SigLIP is also a MI bound. Zhai et al. (2023b) has a discussion on the importance of having the bias term ($b$) in the practical setting to alleviate the imbalance effect of negatives in the initial optimization steps. However, it would be of a future interest to see whether the generalization of SigLIP by either arithmetic or geometric aggregation function to poly-view setting would help to remove the bias term.

### E.4 SUFFICIENT STATISTICS EXTENSION

So far, we have assumed that there is generative factor $\mathbf{c}$ affecting the samples. However, in a more general case, we have multiple factors affecting the sample generation. Let us consider the causal graph presented in Figure 4. Here, we assume that the main factors important for the down-stream task are denoted by $\mathbf{c}$, called *content*, while the non-related factors are shown by $\mathbf{s}_\alpha$, called *styles*. The styles can be different among views while the task-related factor $\mathbf{c}$ is common among them all.

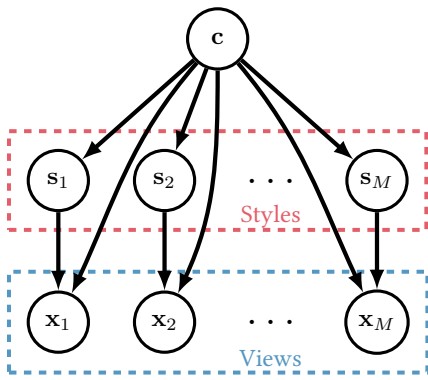

Figure 4: *Content-Style causal graph* A more general poly-view sample generation with task-related generative factor $\mathbf{c}$, called *content*. For each $\alpha \in [M]$, $\mathbf{s}_\alpha$ shows the view-dependent and task non-related factors, called *styles*. The views are shown as before by $\mathbf{x}_\alpha$. In the most general case, content and styles are not independent, while in some settings they might be independent. In the independent scenario, the arrows from $\mathbf{c}$ to $\mathbf{s}_\alpha$ can be ignored.

The goal of this section is to generalize the approach of sufficient statistics introduced in Section 2.4 to the case of content-style causal graph. We start with assuming that content and style are independent and then move to the general case of dependent factors.

**Independent content and style** In this scenario, the arrows in Figure 4 from $\mathbf{c}$ to $\mathbf{s}_\alpha$ will be ignored as there is no dependency between these two factors. We show that for any $\alpha \in [M]$, the sufficient statistics of $\mathbf{x}_\alpha$ with respect to $\{\mathbf{c}, \mathbf{s}_\alpha\}$ has tight relations with sufficient statistics of $\mathbf{x}_\alpha$ to $\mathbf{c}$ and $\mathbf{s}_\alpha$ separately.

**Theorem E.2.** *In the causal generative graph of Figure 4, if $\mathbf{c} \perp\!\!\!\perp \mathbf{s}_\alpha$, then we have:*

$$T_{\{\mathbf{c}, \mathbf{s}_\alpha\}}(\mathbf{x}) = (T_{\mathbf{c}}(\mathbf{x}), \ T_{\mathbf{s}_\alpha}(\mathbf{x})) \quad (125)$$

*Proof.* Having independent factors means $p(\mathbf{x}_\alpha|\mathbf{c}, \mathbf{s}_\alpha) = p(\mathbf{x}_\alpha|\mathbf{c})p(\mathbf{x}_\alpha|\mathbf{s}_\alpha)$. Now, using the Neyman-Fisher factorization (Halmos & Savage, 1949) for sufficient statistics, alongside assuming that we have exponential distribution families, similar to Section 2.4, we have the following

factorization:

$$p(\mathbf{x}_\alpha | \mathbf{c}, \mathbf{s}_\alpha) = R(\mathbf{x}) C(\mathbf{c}, \mathbf{s}_\alpha) \exp\left(T_{\{\mathbf{c}, \mathbf{s}_\alpha\}}(\mathbf{x}) \cdot Q_{\{\mathbf{c}, \mathbf{s}_\alpha\}}(\mathbf{c}, \mathbf{s}_\alpha)\right) \tag{126}$$

$$p(\mathbf{x}_\alpha | \mathbf{c}) = r_1(\mathbf{x}) c_1(\mathbf{c}) \exp\left(T_{\mathbf{c}}(\mathbf{x}) \cdot Q_{\mathbf{c}}(\mathbf{c})\right) \tag{127}$$

$$p(\mathbf{x}_\alpha | \mathbf{s}_\alpha) = r_2(\mathbf{x}) c_2(\mathbf{s}_\alpha) \exp\left(T_{\mathbf{s}_\alpha}(\mathbf{x}) \cdot Q_{\mathbf{s}_\alpha}(\mathbf{s}_\alpha)\right). \tag{128}$$

Substituting these factorizations in the definition of independent generative factors results in:

$$p(\mathbf{x}_\alpha | \mathbf{c}, \mathbf{s}_\alpha) = p(\mathbf{x}_\alpha | \mathbf{c}) p(\mathbf{x}_\alpha | \mathbf{s}_\alpha) \tag{129}$$

$$= (r_1(\mathbf{x}) c_1(\mathbf{c}) \exp\left(T_{\mathbf{c}}(\mathbf{x}) \cdot Q_{\mathbf{c}}(\mathbf{c})\right)) (r_2(\mathbf{x}) c_2(\mathbf{s}_\alpha) \exp\left(T_{\mathbf{s}_\alpha}(\mathbf{x}) \cdot Q_{\mathbf{s}_\alpha}(\mathbf{s}_\alpha)\right)) \tag{130}$$

$$= R(\mathbf{x}) C(\mathbf{c}, \mathbf{s}_\alpha) \exp\left((T_{\mathbf{c}}(\mathbf{x}), T_{\mathbf{s}_\alpha}(\mathbf{x})) \cdot (Q_{\mathbf{c}}(\mathbf{c}), Q_{\mathbf{s}_\alpha}(\mathbf{s}_\alpha))\right), \tag{131}$$

which completes the proof by showing:

$$T_{\{\mathbf{c}, \mathbf{s}_\alpha\}}(\mathbf{x}) = (T_{\mathbf{c}}(\mathbf{x}), \ T_{\mathbf{s}_\alpha}(\mathbf{x})) \tag{132}$$

$$Q_{\{\mathbf{c}, \mathbf{s}_\alpha\}}(\mathbf{c}, \mathbf{s}_\alpha) = (Q_{\mathbf{c}}(\mathbf{c}), \ Q_{\mathbf{s}_\alpha}(\mathbf{s}_\alpha)) \tag{133}$$

$$\square$$

Note that Theorem E.2 also shows that if the space of generative factor $\mathbf{c}$ in Figure 1b is a disentangled space of two or more spaces, i.e. $\mathcal{C} = \mathcal{C}_1 \otimes \mathcal{C}_2$, then the sufficient statistics of $\mathbf{x}$ with respect to $\mathbf{c}$ is equal to a concatenation of sufficient statistics of $\mathbf{x}$ with respect to $\mathbf{c}_1$ and $\mathbf{c}_2$, i.e. sufficient statistics keeps the disentanglement.

**Dependent content and style** When the factors are dependent, the sufficient statistics becomes an entangled measure of $\mathbf{c}$ and $\mathbf{s}_\alpha$. However, if we assume $\mathbf{x}_\alpha = f(\mathbf{c}, \mathbf{s}_\alpha)$ for any $\alpha \in [M]$ and a specific function $f : \mathcal{C} \times \mathcal{S} \mapsto \mathcal{X}$, we have the following theorem:

**Theorem E.3.** *In the causal generative graph of Figure 4, assume for any $\alpha \in [M]$, $\mathbf{x}_\alpha = f(\mathbf{c}, \mathbf{s}_\alpha)$ for an unknown invertible function $f$, such that*

$$\mathbf{c} = f^{-1}(\mathbf{x}_\alpha)_{n_{\mathbf{c}}} \qquad \mathbf{s}_\alpha = f^{-1}(\mathbf{x}_\alpha)_{n_{\mathbf{s}}}, \tag{134}$$

*where $n_{\mathbf{c}}$ and $n_{\mathbf{s}}$ show the elements of $f^{-1}(\mathbf{x}_\alpha)$ that is corresponded to $\mathbf{c}$ and $\mathbf{s}_\alpha$ respectively. Then,*

$$T_{\mathbf{c}}(\mathbf{x}_\alpha) = T_{\mathbf{c}}(\mathbf{x}_\beta) \qquad \forall \, \alpha, \beta \in [M]. \tag{135}$$

*Proof.* Assume that $\mathbf{s}_\alpha$ and $\mathbf{s}_\beta$ are sampled i.i.d from $p_{\mathbf{s}|\mathbf{c}}$. Then, we have:

$$p(\mathbf{x} = \boldsymbol{x}_\alpha | \mathbf{c}) = p(f(\mathbf{c}, \mathbf{s}) = \boldsymbol{x}_\alpha | \mathbf{c}) \tag{136}$$

$$= \delta_{\mathbf{c}} \, p(\mathbf{s} = \boldsymbol{s}_\alpha = f^{-1}(\boldsymbol{x}_\alpha)_{n_{\mathbf{s}}} | \mathbf{c}) \tag{137}$$

$$= \delta_{\mathbf{c}} \, p(\mathbf{s} = \boldsymbol{s}_\beta | \mathbf{c}) \tag{138}$$

$$= \delta_{\mathbf{c}} \, p(\mathbf{s} = \boldsymbol{s}_\beta = f^{-1}(\boldsymbol{x}_\beta)_{n_{\mathbf{s}}} | \mathbf{c}) \tag{139}$$

$$= p(f(\mathbf{c}, \mathbf{s}) = \boldsymbol{x}_\beta | \mathbf{c}) \tag{140}$$

$$= p(\mathbf{x} = \boldsymbol{x}_\beta | \mathbf{c}) \tag{141}$$

Thus, $p(\mathbf{x}_\alpha | \mathbf{c}) = p(\mathbf{x}_\beta | \mathbf{c})$, which using the Neyman-Fisher factorization and exponential family distribution, results in $T_{\mathbf{c}}(\mathbf{x}_\alpha) = T_{\mathbf{c}}(\mathbf{x}_\beta)$. This completes the proof. $\square$

Note that the result of Theorem E.3 recovers the result of von Kügelgen et al. (2021) using the idea of sufficient statistics.

### E.5 OPTIMAL MULTIPLICITY IN THE FIXED BATCH SETTING

In the previous results of Arithmetic and Geometric PVC (Theorem 2.2), we assumed that $M$ can be any number, and accordingly the total number of views $B = K \times M$ for a fixed $K$ increases if $M$ increases. However, it is interesting to investigate the Fixed Batch scenario outlined in Section 3.2 which corresponds to holding $B$ fixed by *reducing* $K$ when $M$ is increased. What is the optimal multiplicity $M^*$ for the bottom row results of Figure 3? We first note that due to the complexity of

MI lower-bounds in Equations 12 and 13, it is not trivial to answer this question as it will depend on the behavior of function $f$ and the map $h$.

Here, we attempt to provide a simplified version of Geometric PVC by adding some assumptions on the behavior of $f$ and $h$. Although this result is for a simplified setting, we believe it provides an interesting insight that for a fixed batch size $B$, depending on the functions $f$ and $h$, there is an optimal number of multiplicity $M^\star$ which maximizes the lower-bound. While even in this simplified version, it is computationally challenging to compute $M^\star$ exactly, it is possible to prove its existence.

In the case that $B$ is fixed, we can rewrite the Geometric PVC in Equation 13 as following by replacing $K = \frac{B}{M}$:

$$\mathcal{I}\left(\mathbf{x}_\alpha; \mathbf{X}_{1:M}^{\neq \alpha}\right) \geq c(B, M) + \mathbb{E}\left[\frac{1}{K}\sum_{i=1}^{K}\frac{1}{M-1}\sum_{\beta \neq \alpha}\log \ell_{i,\alpha,\beta}\right] \tag{142}$$

$$= c(B, M) + \mathbb{E}\left[\frac{M}{B}\sum_{i=1}^{\frac{B}{M}}\frac{1}{M-1}\sum_{\beta \neq \alpha}\log \ell_{i,\alpha,\beta}\right] \tag{143}$$

$$= \mathcal{I}_{\text{Geometric}} \tag{144}$$

To prove that there is an optimal $M$, we need to show that there is $M^\star$ such that $\frac{\partial \mathcal{I}_{\text{Geometric}}}{\partial M} = 0$ at point $M = M^\star$. Since $\ell_{i,\alpha,\beta}$ depends on functions $f$ and $h$ (see Equation 14), and these functions are in practice trained, we assume that for a long enough training of their corresponding neural networks, $\ell_{i,\alpha,\beta}$ converges to its optimum value denoted by $\ell_{i,\alpha,\beta}^\star$. Moreover, we assume that negative samples are uniformly distributed on the hypersphere (following the Wang & Isola (2020) benefits of uniformity criteria). Also, we assume that the convergence of $\ell_{i,\alpha,\beta}^\star$ to its desired value of $1$[5], happens when $M \to \infty$ in a linear way. Note that the first part of this assumption is not limiting since we know as $M$ grows, the lower bound becomes tighter. Due to the fact that one point is $M \to \infty$, there will be many mappings that follow the desired behavior of converging to one as $M$ grows. Here, we introduce two choices for the mapping $\ell_{i,\alpha,\beta}^\star$ with the correct limiting behavior in $M$ to investigate the importance of mapping choice on the optimum value of $M$:

1.  $$\ell_{i,\alpha,\beta}^\star(M) = \boldsymbol{p}^\star + \frac{M-2}{M}(1 - \boldsymbol{p}^\star) = \ell^\star(M), \tag{145}$$

2.  $$\ell_{i,\alpha,\beta}^\star(M) = 1 - \frac{1 - \boldsymbol{p}^\star}{M-1} = \ell^\star(M), \tag{146}$$

where in both equations, $\boldsymbol{p}^\star = \ell_{i,\alpha,\beta}^\star(M = 2)$ and $\lim_{M \to \infty} \ell_{i,\alpha,\beta}^\star(M) = 1$. Note that the uniformity assumption helps to have the same value of $\boldsymbol{p}^\star$ for any $\alpha$ and $\beta$. Now, we can rewrite the $\mathcal{I}_{\text{Geometric}}$ as following:

$$\mathcal{I}_{\text{Geometric}} = c(B, M) + \mathbb{E}\left[\frac{M}{B}\sum_{i=1}^{\frac{B}{M}}\frac{1}{M-1}\sum_{\beta \neq \alpha}\log \ell^\star(M)\right] \tag{147}$$

$$= c(B, M) + \log \ell^\star(M). \tag{148}$$

Now, we can compute $\frac{\partial \mathcal{I}_{\text{Geometric}}}{\partial M} = 0$. We have,

$$\frac{\partial \mathcal{I}_{\text{Geometric}}}{\partial M} = \frac{\partial c(B, M)}{\partial M} + \frac{1}{\ell^\star(M)}\frac{\partial \ell^\star(M)}{\partial M} \tag{149}$$

$$= -\frac{1}{B - M + 1} + \frac{1}{\ell^\star(M)}\frac{\partial \ell^\star(M)}{\partial M} \tag{150}$$

$$= 0. \tag{151}$$

---

[5]The desired value of $\ell_{i,\alpha,\beta}$ is one as it is a likelihood.

Therefore, finding the solution of $\frac{\partial \ell^\star(M)}{\partial M} = \frac{\ell^\star(M)}{B-M+1}$ would provide us with $M^\star$. For each of choices of $\ell^\star(M)$ in Equations 145 and 146, we have:

$$1. \quad \frac{2}{M^\star}(1-\mathrm{p}^\star) - \frac{M^\star - 2(1-\mathrm{p}^\star)}{B - M^\star + 1} = 0 \tag{152}$$

$$2. \quad \frac{1-\mathrm{p}^\star}{M^\star - 1} - \frac{(M^\star - 1) - (1-\mathrm{p}^\star)}{B - M^\star + 1} = 0. \tag{153}$$

Solving these equations results in the following optimum value for each:

$$1. \quad M^\star = \sqrt{2(B+1)(1-\mathrm{p}^\star)} \tag{154}$$

$$2. \quad M^\star = 1 + \sqrt{B(1-\mathrm{p}^\star)}. \tag{155}$$

It is interesting to see that $\mathrm{p}^\star$ plays a role in the optimum value of $M$, which shows the importance of the design of scalar function $f$ and the map $h$. The differences between the values of $M^\star$ in the two scenarios also emphasize the importance of the behavior of the contrastive loss as $M$ increases.

### E.6 SYNTHETIC 1D GAUSSIAN

Following the synthetic setting in Section 3.1, here, we show the following results:

- Provide the proof for the closed form of one-vs-rest MI.
- Evaluate the convergence of the conditional distribution for large $M$ to the true distribution.

**Closed form of one-vs-rest MI** We start with finding the closed form for joint distributions, $p(\mathbf{X}_{1:M})$ and $p(\mathbf{x}_\alpha)p(\mathbf{X}_{1:M}^{\neq\alpha})$. Since all the samples and their views are Gaussian, the joint distribution will also be Gaussian. Thus, it is enough to find the covariance matrix of each density function; note that the mean is set to zero for simplicity.

$$\mathbb{E}\left[\mathbf{x}_\alpha\right] = \int p(\mathbf{c}=\boldsymbol{c})\mathbb{E}\left[\mathbf{x}_\alpha|\mathbf{c}=\boldsymbol{c}\right]\,\mathrm{d}\boldsymbol{c} = \int \boldsymbol{c}p(\mathbf{c}=\boldsymbol{c})\,\mathrm{d}\boldsymbol{c} = \mathbb{E}[\mathbf{c}] = 0, \tag{156}$$

$$\mathrm{Var}\left[\mathbf{x}_\alpha\right] = \mathbb{E}\left[\mathbf{x}_\alpha^2\right] = \int p(\mathbf{c}=\boldsymbol{c})\mathbb{E}\left[\mathbf{x}_\alpha^2|\mathbf{c}=\boldsymbol{c}\right]\,\mathrm{d}\boldsymbol{c} = \sigma^2 + \int \boldsymbol{c}^2 p(\mathbf{c}=\boldsymbol{c})\,\mathrm{d}\boldsymbol{c} = \sigma^2 + \sigma_0^2, \tag{157}$$

$$\mathrm{Cov}\left(\mathbf{x}_\alpha, \mathbf{x}_\beta\right) = \mathbb{E}\left[\mathbf{x}_\alpha\mathbf{x}_\beta\right] = \int p(\mathbf{c}=\boldsymbol{c})\mathbb{E}\left[\mathbf{x}_\alpha\mathbf{x}_\beta|\mathbf{c}=\boldsymbol{c}\right]\,\mathrm{d}\boldsymbol{c} \tag{158}$$

$$= \int p(\mathbf{c}=\boldsymbol{c})\mathbb{E}\left[\mathbf{x}_\alpha|\mathbf{c}=\boldsymbol{c}\right]\mathbb{E}\left[\mathbf{x}_\beta|\mathbf{c}=\boldsymbol{c}\right]\,\mathrm{d}\boldsymbol{c} = \int \boldsymbol{c}^2 p(\mathbf{c}=\boldsymbol{c})\,\mathrm{d}\boldsymbol{c} = \sigma_0^2. \tag{159}$$

Thus, if we present the covariance matrices of $p(\mathbf{X}_{1:M})$ and $p(\mathbf{x}_\alpha)p(\mathbf{X}_{1:M}^{\neq\alpha})$ by $\Sigma_M$ and $\widetilde{\Sigma}_M$ respectively, we have the following:

$$\Sigma_M = \begin{bmatrix} \sigma^2 + \sigma_0^2 & \sigma_0^2 & \cdots & \sigma_0^2 \\ \sigma_0^2 & \sigma^2 + \sigma_0^2 & \cdots & \sigma_0^2 \\ \vdots & \vdots & \ddots & \vdots \\ \sigma_0^2 & \sigma_0^2 & \cdots & \sigma^2 + \sigma_0^2 \end{bmatrix}, \qquad \widetilde{\Sigma}_M = \begin{bmatrix} \sigma^2 + \sigma_0^2 & \mathbf{0} \\ \mathbf{0} & \Sigma_{M-1} \end{bmatrix}. \tag{160}$$

Consequently, we can write the density functions as follows:

$$p(\mathbf{X}_{1:M}) = (2\pi)^{-\frac{M}{2}} \det\left(\Sigma_M\right)^{-\frac{1}{2}} \exp\left(\frac{1}{2}\boldsymbol{x}^\mathsf{T}\Sigma_M^{-1}\boldsymbol{x}\right), \tag{161}$$

$$p(\mathbf{x}_\alpha)p(\mathbf{X}_{1:M}^{\neq\alpha}) = (2\pi)^{-\frac{M}{2}} \det\left(\widetilde{\Sigma}_M\right)^{-\frac{1}{2}} \exp\left(\frac{1}{2}\boldsymbol{x}^\mathsf{T}\widetilde{\Sigma}_M^{-1}\boldsymbol{x}\right), \tag{162}$$

where $\boldsymbol{x} = (x_1, \ldots, x_M)$. Now, we can compute the closed form for one-vs-rest MI using the following lemma:

**Lemma E.4.** *Assume* $\mathbf{x}$ *and* $\mathbf{y}$ *are two multivariate Gaussian random variables of size* $n$ *with laws* $\mathcal{N}_x$ *and* $\mathcal{N}_y$, *covariance matrices* $\Sigma_x$ *and* $\Sigma_y$, *and mean vectors* $\mu_x$, $\mu_y$, *respectively. Then,*

$$\mathcal{D}_{\mathrm{KL}}\left(\mathcal{N}_x \,\|\, \mathcal{N}_y\right) = \frac{1}{2}\left(\mathrm{tr}\left(\Sigma_y^{-1}\Sigma_x\right) + (\mu_y - \mu_x)^\mathsf{T}\Sigma_Y^{-1}(\mu_y - \mu_x) - n + \log\left(\frac{\det\left(\Sigma_y\right)}{\det\left(\Sigma_x\right)}\right)\right). \tag{163}$$

Therefore, the one-vs-rest MI will be as follows:

$$\mathcal{I}(\mathbf{x}_\alpha; \mathbf{X}_{1:M}^{\neq\alpha}) = \mathcal{D}_{\text{KL}}\left(p(\mathbf{X}_{1:M}) \parallel p(\mathbf{x}_\alpha)p(\mathbf{X}_{1:M}^{\neq\alpha})\right) \tag{164}$$

$$= \frac{1}{2}\left(\text{tr}\left(\widetilde{\Sigma}_M^{-1}\Sigma_M\right) - M + \log\left(\frac{\det\left(\widetilde{\Sigma}_M\right)}{\det\left(\Sigma_M\right)}\right)\right). \tag{165}$$

Define $A = \sigma^2 I$, $B = \sigma_0^2 \mathbf{1}$ and $C = A + B$. One can easily see that $A$ and $B$ commute; therefore, they are simultaneously diagonalizable. Thus, there exists matrix $P$ such that the following holds:

$$A = PD_AP^{-1}, \qquad B = PD_BP^{-1}, \qquad C = P(D_A + D_B)P^{-1}, \tag{166}$$

where $D_A$ and $D_B$ show the diagonalized matrices. We also know that $\det(C) = \prod_{i=1}^M \lambda_i(D_A + D_B)$, where $\lambda_i$ denotes the $i$-th eigen value of matrix $D_A + D_B$. Due to the structure of the matrices $A$ and $B$, we can show $D_A = \sigma^2 I$, and $D_B$ is as follows:

$$D_B = \sigma_0^2 \begin{bmatrix} M & 0 & \dots & 0 \\ 0 & 0 & \dots & 0 \\ \vdots & \vdots & \ddots & \vdots \\ 0 & 0 & \dots & 0 \end{bmatrix}. \tag{167}$$

As a result we have the following:

$$\det\left(\Sigma_M\right) = \left(\sigma^2\right)^{M-1}\left(\sigma^2 + M\sigma_0^2\right) \tag{168}$$

$$\det\left(\widetilde{\Sigma}_M\right) = \left(\sigma^2 + \sigma_0^2\right)\det\left(\Sigma_{M-1}\right) = \left(\sigma^2 + \sigma_0^2\right)\left(\sigma^2\right)^{M-2}\left(\sigma^2 + (M-1)\sigma_0^2\right) \tag{169}$$

Also, $\widetilde{\Sigma}_M^{-1}\Sigma_M$ has a block matrix multiplication form:

$$\widetilde{\Sigma}_M^{-1}\Sigma_M = \left(\begin{bmatrix} \sigma^2 + \sigma_0^2 & \mathbf{0} \\ \mathbf{0} & \Sigma_{M-1} \end{bmatrix}\right)^{-1}\begin{bmatrix} \sigma^2 + \sigma_0^2 & \mathbf{v} \\ \mathbf{v}^{\text{T}} & \Sigma_{M-1} \end{bmatrix} \tag{170}$$

$$= \begin{bmatrix} (\sigma^2 + \sigma_0^2)^{-1} & \mathbf{0} \\ \mathbf{0} & \Sigma_{M-1}^{-1} \end{bmatrix}\begin{bmatrix} \sigma^2 + \sigma_0^2 & \mathbf{v} \\ \mathbf{v}^{\text{T}} & \Sigma_{M-1} \end{bmatrix} \tag{171}$$

$$= \begin{bmatrix} 1 & (\sigma^2 + \sigma_0^2)^{-1}\mathbf{v} \\ \Sigma_{M-1}^{-1}\mathbf{v}^{\text{T}} & I \end{bmatrix} \tag{172}$$

Where $\mathbf{v} = (\sigma_0^2, \dots, \sigma_0^2)$ is a $1 \times (M-1)$ matrix. Therefore, $\text{tr}\left(\widetilde{\Sigma}_M^{-1}\Sigma_M\right) = M$. Thus, for the one-vs-rest MI, we have:

$$\mathcal{I}(\mathbf{x}_\alpha; \mathbf{X}_{1:M}^{\neq\alpha}) = \frac{1}{2}\log\left(\left(1 + \frac{\sigma_0^2}{\sigma^2}\right)\left(1 - \frac{\sigma_0^2}{\sigma^2 + M\sigma_0^2}\right)\right) \tag{173}$$

**Convergence of conditional distribution** Using the covariance matrices in Equation 160, and expanding the distributions $p(\mathbf{X}_{i,1:M})$ and $p(\mathbf{X}_{i,1:M}^{\neq\alpha})$, we can write the conditional distribution $p_{\mathbf{x}_{i,\alpha}|\mathbf{X}_{i,1:M}^{\neq\alpha}}$ as follows, which helps us to evaluate Equation 19:

$$p_{\mathbf{x}_{i,\alpha}|\mathbf{X}_{i,1:M}^{\neq\alpha}} = \frac{1}{\sqrt{2\pi\sigma^2}}\left(1 - \frac{\sigma_0^2}{\sigma^2 + M\sigma_0^2}\right)\exp\left[-\frac{\sum_{\alpha=1}^M(\boldsymbol{x}_{i,\alpha} - \bar{\boldsymbol{x}}_i)^2}{2\sigma^2} - \frac{M\bar{\boldsymbol{x}}_i^2}{2(\sigma^2 + M\sigma_0^2)}\right.$$
$$\left. + \frac{\sum_{\beta\neq\alpha}^M(\boldsymbol{x}_{i,\beta} - \bar{\boldsymbol{x}}_i^{\neq\alpha})^2}{2\sigma^2} + \frac{(M-1)(\bar{\boldsymbol{x}}_i^{\neq\alpha})^2}{2(\sigma^2 + (M-1)\sigma_0^2)}\right], \tag{174}$$

where $\bar{\boldsymbol{x}}_i$ and $\bar{\boldsymbol{x}}_i^{\neq\alpha}$ are the average of $\mathbf{X}_{i,1:M}$ and $\mathbf{X}_{i,1:M}^{\neq\alpha}$, respectively. As $M$ increases, $\frac{M\bar{\boldsymbol{x}}_i^2}{2(\sigma^2 + M\sigma_0^2)}$ and $\frac{(M-1)(\bar{\boldsymbol{x}}_i^{\neq\alpha})^2}{2(\sigma^2 + (M-1)\sigma_0^2)}$ converge to $\frac{c_i^2}{2\sigma_0^2}$, which results in these terms cancelling each other. Therefore, we have:

$$\lim_{M\to\infty} p_{\mathbf{x}_{i,\alpha}|\mathbf{X}_{i,1:M}^{\neq\alpha}} = \lim_{M\to\infty}\frac{1}{\sqrt{2\pi\sigma^2}}\exp\left[-\frac{(\boldsymbol{x}_{i,\alpha} - \bar{\boldsymbol{x}}_i)^2}{2\sigma^2}\right.$$
$$\left. + \frac{\sum_{\beta\neq\alpha}(\boldsymbol{x}_{i,\beta} - \bar{\boldsymbol{x}}_i^{\neq\alpha})^2 - (\boldsymbol{x}_{i,\beta} - \bar{\boldsymbol{x}}_i)^2}{2\sigma^2}\right]. \tag{175}$$

Table 1: Hyperparameters for all ImageNet1k experiments in Section 3.2

|  | ResNet 50 |
|---|---|
| Weight initialization | `kaiming_uniform` (He et al., 2015) |
| Backbone normalization | BatchNorm |
| Head normalization | BatchNorm |
| Synchronized BatchNorm over replicas | Yes |
| Learning rate schedule | Single Cycle Cosine |
| Learning rate warmup (epochs) | 10 |
| Learning rate base value | $0.2 \times \frac{4096}{256} = 3.2$ |
| Learning rate minimum value | 0 |
| Optimizer | LARS (You et al., 2017) |
| Optimizer scaling rule | None |
| Optimizer momentum | 0.9 |
| Gradient clipping | None |
| Weight decay | $1 \times 10^{-4}$ |
| Weight decay scaling rule | None |
| Weight decay skip bias | Yes |
| Numerical precision | `bf16` |
| Augmentation stack | `SimCLR` (Chen et al., 2020a) |

Using the central limit theorem, $\bar{\boldsymbol{x}}_i$ converges to $\mathbf{c}_i$, and the second term converges to zero, yielding

$$\lim_{M \to \infty} p_{\mathbf{x}_{i,\alpha} | \mathbf{X}_{i,1:M}^{\neq \alpha}} = p_{\mathbf{x}_{i,\alpha} | \mathbf{c}_i}. \tag{176}$$

## F  REAL-WORLD IMAGE REPRESENTATION LEARNING

### F.1  EXPERIMENTAL DETAILS

**Hyperparameters**  We present the base for training SimCLR (Chen et al., 2020a) and other multi-view methods with a ResNet 50 (He et al., 2016) in Table 1.

**Augmentations**  We use SimCLR augmentations throughout (Chen et al., 2020a), with `color_jitter_strength = 1.0` and an image size override of $224 \times 224$. For completeness, we provide our training augmentation here, our testing augmentation is the standard resize, center crop and normalize, and general multiplicity $M$ corresponds to sampling $M$ independent transformations.

```
[
    transforms.RandomResizedCrop(
        image_size_override, scale=crop_scale, interpolation=Image.BICUBIC
    ),
    transforms.RandomHorizontalFlip(p=0.5),
    transforms.RandomApply(
        [
            transforms.ColorJitter(
                brightness=0.8 * color_jitter_strength,
                contrast=0.8 * color_jitter_strength,
                saturation=0.8 * color_jitter_strength,
                hue=0.2 * color_jitter_strength,
            )
        ],
        p=0.8,
    ),
    transforms.RandomGrayscale(p=0.2),
    transforms.RandomApply([M.GaussianBlur([0.1, 2.0])], p=0.5),
    transforms.ToTensor(),
    IMAGENET_NORMALIZE,
]
```

**Data**  All experiments in Section 3.2 are performed on ImageNet1k (Russakovsky et al., 2014). This dataset is commonly used in computer vision and contains 1.28M training, 50K validation and 100K test images of varying resolutions, each with a label from one of 1000 object classes.

Table 2: Hyperparameters for fine tuning experiments

|  | ResNet 50 |
|---|---|
| Head weight initialization | 0 (Beyer et al., 2022) |
| Head bias initialization | $\ln n_{\text{classes}}^{-1}$ (Beyer et al., 2022) |
| Synchronized BatchNorm over replicas | No |
| Learning rate schedule | Single Cycle Cosine |
| Learning rate warmup (epochs) | 5 |
| Learning rate base value | $\{0.0001, 0.001, 0.001\}$ |
| Learning rate minimum value | 0 |
| Batch size | $\{384, 1024\}$ |
| Training epochs | $\{300, 1000, 2000, 4000\}$ |
| Optimizer | SGD |
| Optimizer scaling rule | SGD |
| Optimizer base batch size | 256 |
| Optimizer momentum | 0.9 |
| Gradient clipping | None |
| Weight decay | 0.0 |
| Numerical precision | `fp32` |
| Augmentation stack | `RandAug` (Cubuk et al., 2020) |
| Repeated Aug. | 2 |
| RandAug | 9/0.5 |
| Mixup prob. | 0.8 |
| Cutmix prob. | 1.0 |
| Random Erasing prob. | 0.25 |

## F.2 FINE-TUNING RESULTS ON TRANSFER TASKS

To investigate whether poly-view methods improve transfer learning performance, we evaluated the ImageNet1k pre-trained models of Section 3.2 by fine-tuning all model weights for a new task.

**Datasets** Following SimCLR (Chen et al., 2020a) we investigated transfer learning performance on the Food-101 dataset (Bossard et al., 2014), CIFAR-10 and CIFAR-100 (Krizhevsky et al., 2014), SUN397 (Xiao et al., 2010), Stanford Cars (Krause et al., 2013), Aircraft (Maji et al., 2013), the Describable Textures Dataset (DTD) (Cimpoi et al., 2014), Oxford-IIIT Pets (Parkhi et al., 2012), and Caltech-101 (Fei-Fei et al., 2007). We report top-1 accuracy for all datasets, and use the predefined train, validation and test splits introduced by the dataset creators.

**Hyperparameters** Fine-tuning hyperparameters are summarized in Table 2. Hyperparameters are optimized for the SimCLR model *only* and are then re-used for the poly-view methods, following the experimental protocol outlined in Section 3.2. All fine-tuning is performed using SGD using momentum. Fine-tuning on smaller datasets is done for a larger number of (e.g. 4k epochs for Aircraft) and with a lower learning rate (e.g. $10^{-4}$ for Caltech-101). Fine-tuning head weights are initialized to zero, with head bias initialized to $\ln n_{\text{classes}}^{-1}$, where $n_{\text{classes}}$ is the number of classes in the corresponding downstream dataset (Beyer et al., 2022).

**Results** In Table 3 we report the test top-1 accuracy after fine-tuning. We do this for the Geometric PVC models with both batch strategies introduced in Section 3.2: *Growing Batch* and *Fixed Batch*, as well as the SimCLR model, for small (256 epochs) and large (1024 epochs) amounts of pretraining. In all cases, Poly-view methods match or outperform the SimCLR baseline for the same set of hyperparameters. We also observe that the Geometric PVC method trained for 256 epochs outperforms the SimCLR method trained for 1024 epochs on transfer to Food, SUN297, Cars, Pets, and Caltech-101, reinforcing the computational efficiency findings of Section 3.2.

| Model | Food | CIFAR10 | CIFAR100 | SUN397 | Cars | Aircraft | DTD | Pets | Caltech-101 |
|---|---|---|---|---|---|---|---|---|---|
| *Pre-training Epochs: 256* | | | | | | | | | |
| Geometric PVC (growing) | **89.56** | **98.33** | 87.38 | **65.91** | 93.47 | 81.13 | 72.41 | 91.31 | 88.27 |
| Geometric PVC (shrinking) | 90.08 | 98.28 | 87.63 | 66.12 | 93.61 | 81.61 | 73.59 | 91.05 | 88.77 |
| SimCLR | 88.58 | 97.76 | **86.55** | **65.10** | 92.96 | 78.54 | 71.44 | 90.51 | 86.49 |
| *Pre-training Epochs: 1024* | | | | | | | | | |
| Geometric PVC (growing) | **90.00** | 98.31 | 87.66 | 66.59 | 93.58 | 80.58 | 73.24 | 90.78 | 89.80 |
| Geometric PVC (shrinking) | 90.27 | 98.41 | 87.73 | 65.68 | 93.78 | 82.42 | 73.30 | 90.88 | 88.57 |
| SimCLR | 89.42 | **98.17** | 86.70 | 65.24 | 93.48 | 80.72 | **72.60** | 90.36 | 89.66 |

Table 3: *Comparison of transfer learning performance of poly-view Geometric PVC against a baseline SimCLR for the same hyperparameter set across nine natural image datasets.* Geometric (growing) and Geometric (shrinking) correspond to Geometric PVC ($M = 8$) using Growing Batch and Shrinking Batch strategies respectively (see Section 3.2). *Top*: ImageNet1k models pre-trained for 256 epochs; *bottom*: ImageNet1k models pre-trained for 1024 epochs. Results not significantly worse than the best (bootstrap confidence interval of 90%) are shown in bold.

### F.3 THE ROLE OF AUGMENTATION STRENGTH AT HIGH MULTIPLICITY

We present the role of augmentation strength at high multiplicity in Figure 5, investigating the effect of different color jittering (Figure 5a) and cropping (Figure 5b). We do not observe significantly different qualitative behavior between the SimCLR baseline and the poly-view methods.

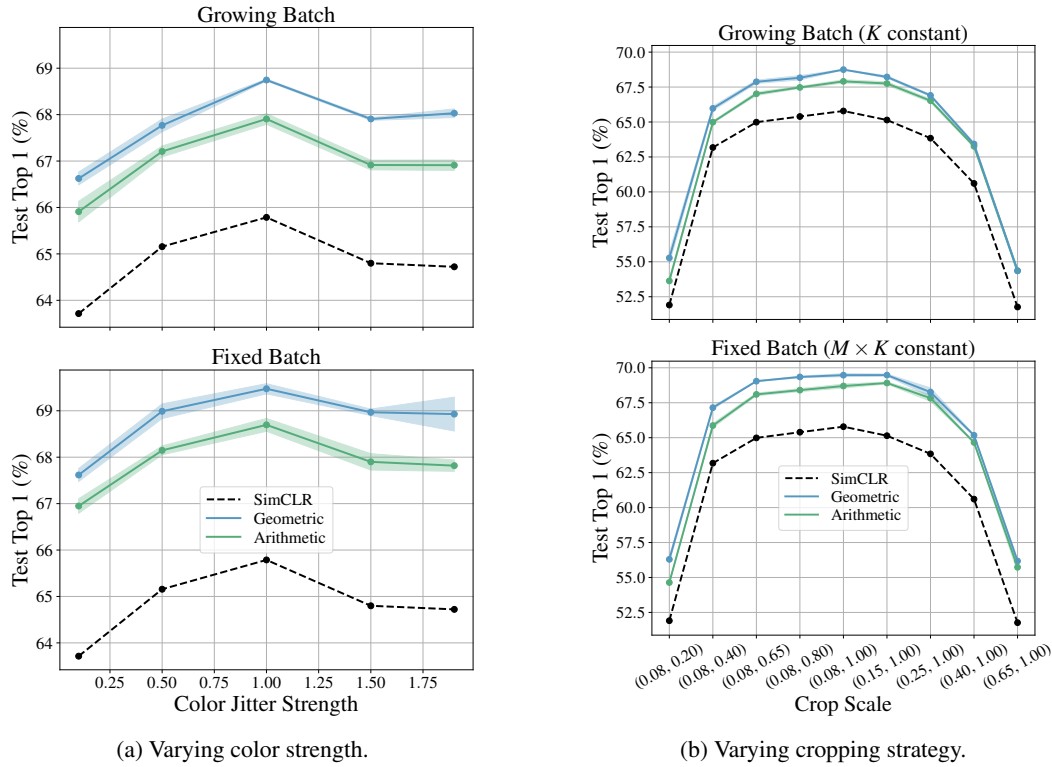

(a) Varying color strength.

(b) Varying cropping strategy.

Figure 5: ResNet 50 trained for 128 epochs with different objectives for different strengths of color augmentation (a) and cropping strategy (b). Geometric and Arithmetic methods presented use multiplicity $M = 4$.

### F.3.1 COMPARISON OF TOTAL FLOATING OPERATIONS

In Section 3.2 and Figure 3a, *Relative Compute* (Equation 29), which measures the total number of encoded views during training, was used to quantify the practical benefits of using Poly-View methods.

To quantify the overall training budget, in Figure 6 we report the total number of FLOPs (FLoating OPerations) performed during training. This is the total number of FLOPs in the forward *and* backward passes for every training step of the model as measured by the PyTorch profiler[6].

The conclusion of Section 3.2 are unchanged when considering total FLOPs instead of Relative Compute. This happens because for sufficiently large models like the ResNet50 we use here, the FLOPs computation is dominated by the model encoder and *not* the loss computation. This results in Relative Compute being a proxy for total FLOPs.

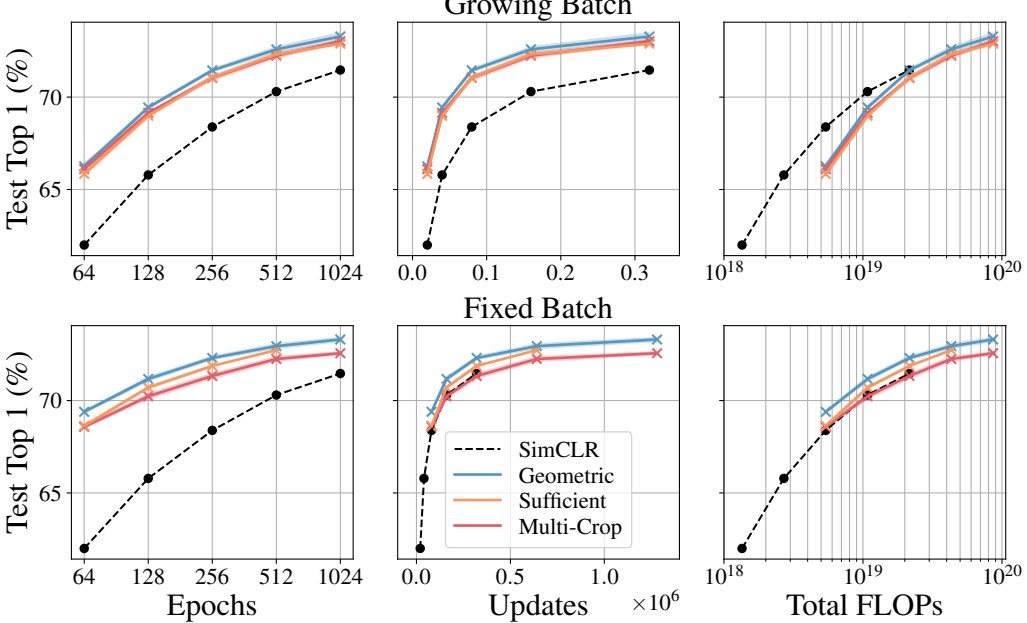

Figure 6: Training at multiplicity $M = 8$ varying training epochs.

Figure 7: *Contrastive ResNet 50 trained on ImageNet1k for different epochs or with different view multiplicities.* Blue, red, orange and black dashed lines represent Geometric, Multi-Crop, Sufficient Statistics, and SimCLR respectively. Bands indicate the mean and standard deviation across three runs. Points indicate *final model performance* of corresponding hyperparameters. We use $K = 4096$ for *Growing Batch* and $K = (2/M) \times 4096$ for *Fixed Batch*. Each method is trained with a multiplicity $M = 8$ except the $M = 2$ SimCLR baseline. We compare models in terms of performance against training epochs (left), total updates (middle) which is affected by batch size $K$, and total FLOPs.

---

[6]https://pytorch.org/docs/stable/profiler.html

### F.3.2 IMPLEMENTATION OF LOSS FUNCTIONS

The pseudocodes for poly-view contrastive loss and sufficient statistics contrastive loss are presented in Algorithms 1 and 2 respectively. Both algorithms have the same structure, with the definition of score matrix as the primary difference. The `rearrange` and `repeat` functions are those of `einops` (Rogozhnikov, 2022).

---

**Algorithm 1** Poly-View Contrastive Loss pseudocode.

---

```
# net            - encoder + projector network
# aug            - augmentation policy
# X[k, h, w, c]  - minibatch of images
# tau            - temperature

def get_mask(beta: int) -> Tensor:
    """The self-supervised target is j=i, beta=alpha. Produce a mask that
    removes the contribution of j=i, beta!=alpha, i.e. return a [k,m,k]
    tensor of zeros with ones on:
    - The self-sample index
    - The betas not equal to alpha
    """
    # mask the sample
    mask_sample = rearrange(diagonal(k), "ka kb -> ka 1 kb")
    # mask the the beta-th view
    mask_beta = rearrange(ones(m), "m -> 1 m 1")
    mask_beta[:, beta] = 0
    return mask_beta * mask_sample                            # [k, m, k]

# generate m views for each sample
X_a = cat([X_1, X_2, ..., X_m], dim=1) = aug(X)               # [k, m, h, w, c]

# extract normalized features for each view
Z = l2_normalize(net(X_a), dim=-1)                            # [k, m, d]

# build score matrix
scores = einsum("jmd,knd->jmnk", Z, Z) / tau                  # [k, m, m, k]

# track the losses for each alpha
losses_alpha = list()

# iterate over alpha and beta
for alpha in range(m):
    losses_alpha_beta = list()
    for beta in range(m):
        # skip on-diagonal terms
        if alpha != beta:
            logits = scores[:, alpha]                         # [k, m, k]
            labels = arange(k) + beta * k                     # [k]
            mask = get_mask(beta)                             # [k, m, k]
            logits = flatten(logits - mask * LARGE_NUM)       # [k, m * k]
            loss_alpha_beta = cross_entropy(logits, labels)   # [k]
            losses_alpha_beta.append(loss_alpha_beta)
    losses_alpha = stack(loss_alpha, dim=-1)                  # [k, m-1]

    # aggregate over the betas for each alpha
    if method == "arithmetic":
        loss_alpha = logsumexp(losses_alpha, dim=-1) - log(k) # [k]
    elif method == "geometric"
        loss_alpha = mean(losses_alpha, dim=-1)               # [k]

    losses_alpha.append(loss_alpha)

# build final loss matrix
losses = stack(losses_alpha, dim=-1)                          # [k,m]

# take expectations
sample_losses = mean(losses, dim=-1)                          # [k]
loss = mean(sample_losses)                                    # scalar
```

---

**Algorithm 2** Sufficient Statistics Contrastive Loss pseudocode.

```
# net           - encoder + projector network
# aug           - augmentation policy
# X[k, h, w, c] - minibatch of images
# tau           - temperature

def get_mask(beta: int) -> Tensor:
    """The self-supervised target is j=i, beta=alpha. Produce a mask that
    removes the contribution of j=i, beta!=alpha, i.e. return a [k,m,k]
    tensor of zeros with ones on:
    - The self-sample index
    - The betas not equal to alpha
    """
    # mask the sample
    mask_sample = rearrange(diagonal(k), "ka kb -> ka 1 kb")
    # mask the the beta-th view
    mask_beta = rearrange(ones(m), "m -> 1 m 1")
    mask_beta[:, beta] = 0
    return mask_beta * mask_sample                              # [k, m, k]

# generate m views for each sample
X_a = cat([X_1, X_2, ..., X_m], dim=1) = aug(X)                # [k, m, h, w, c]

# extract normalized features for each view
Z = l2_normalize(net(X_a), dim=-1)                             # [k, m, d]

# build the average of the rest-set
# step 1: repeat the features M times
Z_tilde = repeat(Z, "k m1 d -> k m1 m2 d", m2=m)               # [k, m, m, d]

# step 2: replace the effect of alpha-th view by zero
# and correct the bias coefficient of mean
diagonal_one = rearrange(eye(m), "m1 m2 -> 1 m1 m2 1")
diagonal_zero = ones([k, m, m, d]) - diagonal_one              # [k, m, m, d]
Z_tilde = m / (m - 1) * Z_tilde * diagonal_zero                # [k, m, m, d]

# step 3: getting the average of rest-set and nomalize
Z_tilde = mean(Z_tilde, dim=1)                                 # [k, m, d]
Z_tilde = l2_normalize(Z_tilde, dim=-1)                        # [k, m, d]

# build score matrix
scores = einsum("jmd,knd->jmnk", Z, Z_tilde) / tau             # [k, m, m, k]

# track the losses for each alpha
losses_alpha = list()

# iterate over alpha and beta
for alpha in range(m):
    losses_alpha_beta = list()
    for beta in range(m):
        # skip non-diagonal terms
        if alpha == beta:
            logits = scores[:, alpha]                          # [k, m, k]
            labels = arange(k) + beta * k                      # [k]
            mask = get_mask(beta)                              # [k, m, k]
            logits = flatten(logits - mask * LARGE_NUM)        # [k, m * k]
            loss_alpha_beta = cross_entropy(logits, labels)    # [k]
            losses_alpha_beta.append(loss_alpha_beta)
    losses_alpha = stack(loss_alpha, dim=-1)                   # [k, m-1]

    # aggregate over the betas for each alpha
    loss_alpha = mean(losses_alpha, dim=-1)                    # [k]

    losses_alpha.append(loss_alpha)

# build final loss matrix
losses = stack(losses_alpha, dim=-1)                           # [k,m]

# take expectations
sample_losses = mean(losses, dim=-1)                           # [k]
loss = mean(sample_losses)                                     # scalar
```

## G EXPANDED RELATED WORK

**SSL methods and contrastive learning**   Contrastive learning appears in many SSL methods. Sim-CLR (Chen & He, 2021) leverages the InfoNCE objective to train the encoders to find good representations. MoCo (He et al., 2019; Chen et al., 2020b; 2021) uses a momentum encoder to create a moving average of the model's parameters, enabling it to learn powerful image representations without the need for labeled data. CLIP (Radford et al., 2021) is a novel multi-modal approach that leverages contrastive learning to bridge the gap between text and images. VICReg (Bardes et al., 2021) is another SSL method that uses contrastive learning but also address the collapse problem in which the encoders produce non-informative vectors using regularization terms. There are some works (Shwartz-Ziv et al., 2023; Balestriero & LeCun, 2022) providing theoretical understanding of VICReg's performance and comparing it to the other methods like SimCLR. Tian et al. (2020a) is the closest work we know of that has investigated a simple form of multiplicity view in contrastive learning. Their approach is to get the average of pairwise contrastive learning, which translates to the lower-bound of Validity property that we have, for which we have shown theoretically in Equation 8 that our aggregation function outperforms this lower-bound. We also note that the authors did not provide any theoretical explanation regarding multiplicity.

**Information-theoretic perspective in SSL**   One of the main approaches to understand SSL and contrastive learning is to study the dependencies between pairs of variables or views. MI provides an insightful metric for quantifying dependencies resulting to the point that estimating and optimizing the MI becomes important. van den Oord et al. (2018) introduces InfoNCE loss for the first time. It combines predicting future observations with a probabilistic contrastive loss, hence the name Contrastive Predictive Coding. The intuition behind this work is that different parts of the signal share same information. Poole et al. (2019) provides a framework to estimate MI by showing connections between different MI lower-bounds, and investigating the bias and variance of their sample-based estimators. Tschannen et al. (2020) leverages this framework and builds connection between MI maximization in representation learning and metric learning by also pinpointing that under which dependency conditions MI approaches perform well. Lee et al. (2023) provides more insights on MI maximization in contrastive learning like the effect of same-class-sampling for augmentations by upper-bounding the MI. Gálvez et al. (2023) shows that not only contrastive SSL methods, but also clustering methods (Caron et al., 2020; 2018), and (partially) distillation methods (Grill et al., 2020; Caron et al., 2021) implicitly maximize MI.

**The role of augmentation in SSL**   Augmentation is a critical part of SSL methods in computer vision to keep the task-relevant information (Tian et al., 2020b). Trivial augmentations result in non-informative representations, preventing the model to find the main features to distinguish between positives and negatives, while hard augmentations make it difficult for the model to classify the positives from negatives. Balestriero et al. (2022b) tackles this problem by quantifying how many augmentations are required for a good sample-based estimation of MI to have low variance and better convergence. Kim et al. (2023) addresses this challenge in contrastive learning with a different approach and by adding weights that implies the goodness of the augmentation. On the importance of augmentation, von Kügelgen et al. (2021) shows that augmentation helps to disentangle the content from the style in the images. From another perspective, some works explore the effect of different augmentation strategy like multi-crop in contrastive learning and SSL methods (Caron et al., 2020; 2021). Fort et al. (2021) has a similar setting to ours and shows that increasing the number of augmentations, i.e. increasing the signal to noise ratio, helps the supervised learning classifier in both growing batch and fixed batch scenarios. Wang & Qi (2022) studied the effect of strong augmentations in contrastive learning, proposing a new framework to transfer knowledge from weak augmentations to stronger ones in order to address the loss of information due to harsh transformations; improving the performance.

**Sufficient Statistics**   Sufficient statistics provide a summary of the data, from which one can make inferences on the parameters of the model without referencing samples. Sufficient statistics can be readily connected to the Infomax principle and have been used to re-formulate contrastive learning Chen et al. (2020c). One key observation is that two-view contrastive learning may not yield representations that are sufficient w.r.t. the information they contain to solve downstream tasks (Tian et al., 2020a; Wang et al., 2022). Poly-view contrastive learning tasks have an increased amount of

available information shared between views, which we believe improves the resulting representations' sufficiency w.r.t. any possible downstream task that would have been possible from the unknown generative factors.

# H EXTENSIONS TO DISTILLATION

Our primary contributions use the frameworks of information theory and sufficient statistics to investigate what is possible in the presence of a view multiplicity $M > 2$ and derive the different Poly-View objectives from first principles.

It is possible to incorporate multiplicity $M > 2$ into a distillation setup like BYOL (Grill et al., 2020). For example, DINOv1 (Caron et al., 2021), which shares many algorithmic parts of BYOL, benefits a lot from using the pair-wise Multi-Crop task that we described in Section 2 and Appendix D (although in DINOv1, there is more than one augmentation policy).

One option for extending distillation methods like BYOL and DINOv1 from Multi-Crop to poly-view tasks in a One-vs-Rest sense is to have the EMA teacher produce $M - 1$ logits, which are aggregated into a single logit (similar to the sufficient statistics choice for $Q$ in Equation 24) for producing the target pseudo-label distribution. The gradient-based student could then be updated based on its predictions from the held-out view, and this procedure aggregated over all the view hold-outs.

The core difference between the distillation procedure above and the poly-view contrastive methods is that the large-view limit of poly-view contrastive methods is provably a proxy for InfoMax (Section 2 and Equation 26). There may be a way to obtain theoretical guarantees for the large-view distillation methods (using for example tools from Gálvez et al. (2023)), and could prove an interesting future direction for investigation.

# I CONTRIBUTIONS

All authors contributed to writing this paper, designing the experiments, and discussing results at every stage of the project. All contributions are in alphabetical order by last name.

**Writing and framing** Majority of writing done by Dan Busbridge, Devon Hjlem and Amitis Shidani. Research framing done by Dan Busbridge and Devon Hjlem.

**Theoretical results** Proofs of MI lower-bound with Multi-Crop (Appendix C.1), lower variance of Multi-Crop MI bound (Appendix C.2), Generalized $\mathcal{I}_{\mathrm{NWJ}}$ (Appendix C.3), Validity of Arithmetic and Geometric PVC (Appendix C.4), the connection between sufficient statistics and MI bounds (Appendix C.7), the generalization of one-vs-rest MI to arbitrary set partitions (Appendix E.1), and the linking of poly-view methods to SimCLR (Appendix E.2) and SigLIP (Appendix E.3) done by Amitis Shidani. Derivation of optimal multiplicity (Appendix E.5) done by Amitis Shidani, Dan Busbridge and Devon Hjelm. Derivation of Arithmetic and Geometric PVC loss functions (Appendix C.5) done by Dan Busbridge and Amitis Shidani. Proof of Behavior of MI Gap (Appendix C.6) done by Eeshan Gunesh Dhekane and Amitis Shidani.

**Sufficient statistics** Extension to sufficient statistics framework (Appendix E.4) proposed by Devon Hjlem. Derivation of sufficient statistics loss function (Equation 22) done by Amitis Shidani and Russ Webb. Derivation of $Q$ (Equation 24) done by Dan Busbridge.

**Synthetic 1D Gaussian** Synthetic setting proposed by Dan Busbridge and Amitis Shidani based on discussions with Devon Hjelm. One-vs-rest MI (Equations 26 and 173) derived by Dan Busbridge and Amitis Shidani. Proofs of convergence to InfoMax (Equation 27) and to the sufficient statistic conditional distribution (Equations 28 and 176) derived by Amitis Shidani. Code to produce empirical results (Figure 2) and related analysis by Amitis Shidani.

**Real world representation learning** Experimental protocol for training duration experiments (Section 3.2 and Appendix F.3.1) designed by Dan Busbridge. Experiments conducted by Dan

Busbridge, Eeshan Gunesh Dhekane, Jason Ramapuram, Amitis Shidani, and Russ Webb. Fine-tuning transfer experiments (Appendix F.2) done by Jason Ramapuram. Investigation into the role of augmentation strength (Appendix F.3) done by Amitis Shidani.

**Implementation details**  ImageNet1k investigations carried out in PyTorch distributed frameworks developed by Dan Busbridge, Eeshan Gunesh Dhekane and Jason Ramapuram. Design and implementation of fast poly-view contrastive losses (Algorithm 1) by Dan Busbridge. Design and implementation of fast sufficient statistics loss (Algorithm 2) by Amitis Shidani. Baseline implementation of SimCLR, by Jason Ramapuram.

