# OpenReview forum: "Poly-View Contrastive Learning"
_ICLR.cc/2024/Conference — ICLR 2024 poster_

### Official Review · Reviewer_PqKH · 2023-10-30

**Soundness:** 3 good
**Presentation:** 3 good
**Contribution:** 3 good
**Rating:** 6
**Confidence:** 2

**Summary:**

This paper studies contrastive learning by matching more than two related views, which is called poly-view contrastive learning. Unlike traditional contrastive learning methods that take pairs of tasks, it increases the view multiplicity and investigates the design of SSL tasks that use many views. Experiments show that it is beneficial to decrease the number of unique samples while increasing the number of views of each sample.

**Strengths:**

The idea of designing contrastive learning methods using poly-view seems novel. It utilizes an observation from the prior works that using multiple positive views improves the performance. The paper is well-written.

**Weaknesses:**

Although there are prior works showing that multiplicity improves generalization and convergence of neural networks, it lacks rigorous theory on the relation between contrastive learnability and the number of views on each sample.

**Questions:**

I wonder how strong the theory on multiplicity can be. Is it possible to specify how exactly the number of views on each example improves the algorithmic performance? Would it be essential on the average number of views, or maximal number? Does there exist a threshold on the number of views, such that once it exceeds the threshold, more views do not help?

---

> ### Author Response · Authors · 2023-11-16
>
> Thank you for taking the time to read our work. We are happy you found the paper well-written and that you appreciate the benefits of being able to trade off batch size against views as well as the additional insights we provide about self-supervised learning algorithm design.
>
> Please find below our comments on the raised issues and questions:
>
> ### *W1/Q1: Although there are prior works showing that multiplicity improves generalization and convergence of neural networks, it lacks rigorous theory on the relation between contrastive learnability and the number of views on each sample. I wonder how strong the theory on multiplicity can be.*
>
> Two-view MI objectives found in prior works on contrastive learning are a proxy to the InfoMax objective for representation learning. Our theory shows that one need not limit themselves to two-view MI to achieve InfoMax, but rather many other alternatives exist that use more positive views. Our paper studies One-vs-Rest MI, multiplicity being a controllable parameter and two-view being the case when the multiplicity $M=2$, yielding the following theoretical and empirical results:
> 1. Using multiple views in a pairwise comparison as in Multi-Crop, while reducing the variance of the MI estimator, does *not* result in a better bound for InfoMax (Proposition 2.1 and 2.2).
> 2. New contrastive losses that include multiple positive views, such as Arithmetic and Geometric PVC provide a tighter lower-bound on InfoMax  (Theorem 2.3).
> 3. Sufficient statistics with minimal assumption also introduce a way of including more positive views into account. For a specific choice of sufficient statistics, we prove a connection between sufficient statistics and One-vs-Rest MI (Theorem 2.4) which shows that sufficient statistics also benefit from the above properties of One-vs-Rest MI.
> 4. Using synthetic and real-world (ImageNet1k) datasets, these theorems were validated in practical settings.
>
> Please provide clarification regarding definitions of:
> - Strength of multiplicity theories, and
> - Contrastive learnability
>
> in the case where we have not addressed your concerns.
>
> ### *Q2: Does there exist a threshold on the number of views, such that once it exceeds the threshold, more views do not help?*
>
> In the Growing Batch case (Figure 3, top row), one should always maximize the multiplicity $M$ as this provides the tightest bound on InfoMax (Theorem 2.3, 2.4).
>
> In the Fixed Batch case where the total number of views $V=K\times M$ is held constant as $M$ increases, the answer is more interesting.
> Prompted by your comment, in Appendix E.5 we investigate this scenario in more detail with a simplified form of Geometric PVC, and are able to prove there does exist an optimal multiplicity $M^*$. Thank you for prompting us with this comment.
>
> Thank you again for the time you took reading our work and providing the questions above. The additions to the work regarding Fixed Batch view multiplicity in particular have improved the work's overall completeness.

---

> > ### Author Response · Authors · 2023-11-20
> > **Request for remaining issues**
> >
> > Dear Reviewer PqKH,
> >
> > We thank you again for your thoughtful comments, and particularly your Q2 regarding the existence of a threshold number of views. We are very happy we were able to provide the requested existence proof as part of this discussion period.
> >
> > Please let us know if you have any other issues remaining with the work.
> >
> > Best,
> >
> > The Authors

---

### Official Review · Reviewer_P4Q6 · 2023-10-31

**Soundness:** 3 good
**Presentation:** 3 good
**Contribution:** 2 fair
**Rating:** 6
**Confidence:** 4

**Summary:**

The paper investigates the effect when introducing view multiplicity in contrastive learning. Specifically, the paper gives a generic information-theoretic derivation of such multi-view framework and shows that SimCLR loss is a special case of the derived `poly-view' contrastive learning. The paper concluded from the theoretical foundation that higher view multiplicity enables a new contrastive learning where, surprisingly, it is beneficial to reduce the batchsize and increase multiplicity. The paper also associate their  theoretical findings with experiments.

**Strengths:**

The paper delves into the impact of incorporating multiple perspectives in contrastive learning. The strength include:

S1: it presents a comprehensive information-theoretic analysis of this multi-view approach and establishes that the SimCLR loss can be considered as a special case of the resulting 'poly-view' contrastive learning.

S2: Based on the theoretical framework, the research concludes that greater view multiplicity facilitates a novel form of contrastive learning, wherein it proves unexpectedly advantageous to decrease the batch size while augmenting the multiplicity.

S3: Furthermore, the paper justifies these theoretical discoveries with empirical experiments.

**Weaknesses:**

However, the paper has several weakness that worths further discussion.

W1: What is the exact loss function the paper used to define the poly-view contrastive learning? It seems the Eq. (22) is the poly-view contrative loss, whereas it is in a very high level abstract and implicit form, making it hard to interprete how to compute the empirical loss for the terms, and why M=2 links to SimCLR. I recommend to make the loss in a more explicit form of empirical losses and interpretation (e.g., what is the used sufficient statistics for M=2 for SimCLR? )

W2: Empirical evidence lacks suitable interpretation and linkage to the significance of the theorems. For significance, I mean how we can use the theorem takeaways to practically improve the SSL algorithms? I expect to see the evidence on larger dataset with mainstream architecture such as ResNet and ViT/transformer with longer epochs.

W3: It is unclear to me if the multiplicity of views simply benefits from more equivalent of epochs (in the experiments) or whether the exposure to the number of data has been constrained to be exactly same between the poly-view contrastive learning and other baselines.

W4: There is no comparison between the proposed method and SOTA method, in terms of how the method contributes to and improves the SOTA methods under the theoretical foundations.

**Questions:**

Please see the 4 weakness above for questions to be addressed.

**Details Of Ethics Concerns:**

None.

---

> ### Author Response · Authors · 2023-11-16
> **Response to Reviewer P4Q6 (1/2)**
>
> Thank you for taking the time to read and review our work. We are happy you appreciated the theoretical analysis of view multiplicity and the empirical justifications of the theoretical analysis.
>
> Please find below our comments on the raised issues and questions:
>
> ### *W1.1: What is the loss for PVC?*
> Minimizing the PVC losses maximizes the MI lower bounds of Theorem 2.2. Based on your comment we have clarified the corresponding equations, making it clear what the loss is. Additionally, the full pseudo-code for all loss computations is provided in Appendix F.2.5, allowing any reader to follow the loss computation line by line and reproduce our results.
>
> ### *W1.2: How does M=2 link to SimCLR?*
> In Appendix E.2 we show how $M=2$ gives SimCLR for PVC. We have now added an equivalent derivation for sufficient statistics.
>
> ### *W2.1: Empirical evidence lacks suitable interpretation and linkage to the significance of the theorems. For significance, I mean how we can use the theorem takeaways to practically improve the SSL algorithms?*
> The PVC and Sufficient Statistic algorithms are a direct consequence of Theorems 2.2 and 2.4 respectively. These theorems tell you how to construct a Poly-View loss function. Without the theorems, one can use heuristics and develop Multi-Crop, which Poly-View methods outperform in all of our experiments.
>
> In the synthetic case (Section 3.1) we observe experiments directly following theorems from Section 2. For example, increasing multiplicity $M$ increases One-vs-Rest MI as derived in Section 2.3.1 on data processing inequality. We also observe One-vs-Rest MI empirically converges to $\mathcal I(x_\alpha; c)$ which, as discussed in Section 2.3.1, is the upper bound for One-vs-Rest MI.
>
> The ImageNet1k experiments follow similarly. Geometric and Sufficient Statistics with higher multiplicity $M$ provide tighter MI bounds and a better proxy for InfoMax, resulting in improved performance when holding other hyperparameters fixed (Figure 3a).
>
> ### *W.2: I expect to see the evidence on larger dataset with mainstream architecture such as ResNet and ViT/transformer with longer epochs*
> Please see Figure 3a for training and evaluation on ImageNet1k dataset, with a ResNet50 architecture trained for up to 1024 epochs.
>
> ### *W.3: It is unclear to me if the multiplicity of views simply benefits from more equivalent epochs (in the experiments) or whether the exposure to the number of data has been constrained to be exactly the same between the poly-view contrastive learning and other baselines*
> We were also concerned about this, and chose to focus on it in Section 3.2.
>
> For Figure 3b, we train $M=2$ SimCLR models and $M=8$ Poly-View for 64, 128, 256, 512 and 1024 epochs. We do this either by holding the total number of views fixed (Fixed Batch. bottom row) which means we drop the batch size $B$ with increasing $M$, or allow total views to increase (Growing Batch, top row).
>
> 1. To answer whether it is the total number of samples seen by the model, we can look at the leftmost column of Figure 3a. We see that Poly-View methods always outperform SimCLR.
> 2. To answer whether it is the total number of updates seen by the model, we can look at the center column of Figure 3a. We see that Poly-View methods always outperform SimCLR.
> 3. To answer whether it is the total compute spent on the model, we can look at the right column of Figure 3a. Poly-View methods outperform SimCLR only in the Fixed Batch case, but not in the Growing Batch.
>
> If *effective epochs* corresponds to *total steps* then in the above scenario 2 we see that for the same amount of effective epochs, Poly-View methods still outperform non-Poly-View methods.
>
> We have also added a fourth way of comparing models in Appendix F.2.4 where we show the Total FLOPs used to produce each model. The conclusion is consistent with comparisons based on Relative Compute.
>
> Please let us know if anything else is still not clear.

---

> > ### Author Response · Authors · 2023-11-16
> > **Response to Reviewer P4Q6 (2/2)**
> >
> > ### *W4: There is no comparison between the proposed method and SOTA method, in terms of how the method contributes to and improves the SOTA methods under the theoretical foundations*
> >
> > The goal of our work is to understand how view multiplicity should be incorporated into contrastive learning from first principles and to investigate in a controlled ImageNet1k setting if there is any benefit to doing so in this way. Achieving state-of-the-art performance is not a goal of our work.
> >
> > We found:
> > - Increasing multiplicity of Poly-View methods always gives better performance than the fixed multiplicity method SimCLR (Fig 3a top row first two columns, Fig 3b).
> > - Poly-View methods enable more compute-efficient design space than fixed multiplicity methods (Fig 3a bottom row final column).
> >
> > This shows that taking a Poly-View approach opens up more compute-efficient self-supervised algorithms. We hope this helps clarify the purpose of our work and the benefits of a Poly-View approach which may be incorporated with orthogonal methods for increasing performance, e.g. increasing model size.
> >
> > Additionally, we have added Appendix H where we discuss how Poly-View approaches can be included in other types of self-supervised learning algorithms, like distillation based methods such as BYOL and DINO.
> >
> > Thank you again for the time you took reading our work and providing valuable comments and questions. The resulting modifications and additions to the work have improved the overall clarity and completeness.

---

> ### Author Response · Authors · 2023-11-20
> **Request for remaining issues**
>
> Dear Reviewer P4Q6,
>
> We thank you again for taking the time to read and review our work. We really appreciated your comments and questions that prompted a number of clarifications and missing section links to be included in the paper, which has improved the overall readability of the paper. In particular, the section discussing the Geometric and Arithmetic PVC loss definitions is now much clearer, thank you.
>
> Please let us know if you have any issues remaining with the work.
>
> Best,
>
> The Authors

---

> > ### Comment · Reviewer_P4Q6 · 2023-11-22
> > **Thanks for the further clarification**
> >
> > As the authors adequately pointed out the concerns in both the theoretical part and empirical part, I have increased my score. I hope the updated version could make the explicit loss and its linkage to SimCLR clearer in the main paper, rather than in the appendix.

---

### Official Review · Reviewer_AhEA · 2023-11-03

**Soundness:** 3 good
**Presentation:** 3 good
**Contribution:** 3 good
**Rating:** 6
**Confidence:** 3

**Summary:**

Although it is possible to design tasks that drawn arbitrary number of views, contrastive works typically focus on pairwise tasks. So, this paper investigates how to match more than two related views in contrastive learning, and derive new learning objectives by information maximization and sufficient statistics. They show that multi-crop reduces the variance of corresponding paired objective but fail to improve bounds on MI; Then they derive new objectives which solve tasks across all views through information theory, and show that the MI Gap is monotonically non-increasing with respect to the number of views. Also, the poly-view contrastive method is beneficial to reduce the batch size and increase multiplicity.

**Strengths:**

1. Generalizing the information-theoretic foundations to poly-view is an interesting idea, and the One-vs-Rest MI seems to be quite reasonable.

2. Those theoretical results are clear, the derivation process of the One-vs-Rest objective is convincing.

3. The paper is well written and easy to follow.

**Weaknesses:**

1. The assumption 3 in section 2.4.1 is kind of strong.

2. The experiments do not show the superiority of poly-view contrastive learning. The computation time is not evaluated by real time, and the downstream performance is not displayed.

**Questions:**

1. The experiments shown in section 3 display the relative compute of algorithms and show One-vs-Rest objectives could beat simCLR with the same relative compute, how about the real training complexity. And how does it perform on real downstream tasks.

2. I cannot fully understand why the One-vs-Rest loss could effectively reduce the training epoch and batch size.

3. The Geometric loss is actually also an extension of simCLR loss, just like Multi-Crop, but Geometric loss could be a tighter bound of MI while Multi-Crop cannot. It seems to be theoretically correct, but how can we understand it empirically.

4. An extension of simCLR loss outperforms the carefully designed SUFFICIENT STATISTICS loss in section 3, does it mean that the poly-view contrastive learning works mainly because the superiority of simCLR loss?

---

> ### Author Response · Authors · 2023-11-16
> **Response to Reviewer AhEA (1/2)**
>
> Thank you for taking the time to read our work. We are happy you found the paper easy to read and you appreciate the benefits of tighter MI bounds, lower batch size, and reduced computational cost enabled by the Poly-View objectives, as well as the additional insight we provide about Multi-Crop.
>
> Please find below our comments on the raised issues and questions:
>
> ### *W1: Exponential family assumption is strong*
>
> Our representations are given by a neural network and are therefore finite-dimensional (2048 in the ResNet50 case), typical in self-supervised learning. As the sufficient statistics is a map to the representation space, according to the Fisher-Darmois-Koopman-Pitman (FDKP) theorem, for smooth nowhere vanishing probability densities, a finite-dimensional sufficient statistic exists if and only if the densities are from an exponential family (https://ieeexplore.ieee.org/document/4048925).
> I.e. the sufficient statistics are distributed according to the exponential family because they map to finite-dimensional representations.
>
> The assumption of finite-dimensional representations is not strong in the context of self-supervised learning.
> The exponential families that follow from this assumption are widely used in machine learning due to the availability of efficient methods as well as their tractability and interpretability. Many traditional distributions are in the exponential family, e.g. normal, exponential, log-normal, gamma, chi-squared, beta, Dirichlet, Bernoulli, categorical, Poisson, geometric, inverse Gaussian, von Mises, and von Mises-Fisher distributions.
> Finally, in self-supervised learning, the exponential families have appeared in many recent works, e.g. https://arxiv.org/abs/2102.08850, https://arxiv.org/abs/2203.07004, and https://proceedings.mlr.press/v89/hyvarinen19a.html.
>
> We have reworded assumption 3 to make it clear it is a consequence of modeling representations using a neural network, and *not* an assumption. Thank you for highlighting this to us.
>
> ### *W2: The experiments do not show the superiority of Poly-View contrastive learning*
>
> The goal of our work is to understand how view multiplicity should be incorporated into contrastive learning from first principles and to investigate in a controlled ImageNet1k setting if there is any benefit to doing so in this way. Achieving state-of-the-art performance is not a goal of our work.
>
> We found:
> - Increasing multiplicity of Poly-View methods always gives better performance than the fixed multiplicity method SimCLR (Fig 3a top row first two columns, Fig 3b)
> - Poly-View methods enable more compute-efficient design space than fixed multiplicity methods (Fig 3a bottom row final column).
>
> We hope this helps clarify the purpose of our work and the benefits of a Poly-View approach which may be incorporated with orthogonal methods for increasing performance, e.g. increasing model size. We will continue to improve the clarity of the paper on this particular point. Please let us know if there is anything else you are looking for.
>
> ### *W3/Q1: Computation time is not evaluated by real time*
>
> In Figure 3 we evaluate computation time using:
> - Total training epochs
> - Total updates
> - Relative Compute
>
> These metrics capture different notions of time, as choosing only one or two of these metrics paints an incomplete picture. E.g., displaying only total updates does not take into account the practical consideration that a single update for one setting may require more computation than a single update for another, which would be captured by also including Relative Compute.
>
> The three metrics above were also chosen as they are ML framework agnostic and hardware agnostic. Wall-clock time on the other hand can vary according to CUDA version, GPU model/compute capability network interconnect (e.g. infiniband/EFA availability). This makes wall-clock time difficult to use reproducibly.
>
> Based on your comment, we have included total FLOPs, another popular, reproducible measure of computation time, in Appendix F.2.4, to complete the picture. We have also added a comment that, assuming an optimal hardware setting and fully-parallel implementation, total updates is proportional to wall-clock time, which we hope addresses your primary concern. Thank you again for your comment.

---

> > ### Author Response · Authors · 2023-11-16
> > **Response to Reviewer AhEA (2/2)**
> >
> > ### *W4/Q1: Downstream performance is not displayed*
> >
> > We are now fine-tuning Poly-View methods on the transfer tasks of Table 8 in https://arxiv.org/abs/2002.05709. Results will be included in Appendix F.2.2 when they are complete.
> > Thank you for suggesting this.
> >
> > ### *Q2: Why does One-vs-Rest reduce 1) training epochs and 2) batch size*
> > 1. One-vs-Rest for high multiplicity $M$ is a better proxy for InfoMax (Section 2.1 and Equation 27), and so increasing $M$ gives a better signal to noise. There is also an optimizer scaling law interpretation (https://arxiv.org/abs/2102.12470, https://arxiv.org/abs/2307.13813) where we can trade 2 steps of 2 views per sample with one step of 4 views per sample.
> > 2. Based on 1., One-vs-Rest means you can maintain performance if you reduce your batch size if you increase $M$ accordingly.
> >
> > ### *Q3: Does PCV outperform sufficient statistics because of superiority of SimCLR loss?*
> > Sufficient statistics with $M=2$ is SimCLR (Section 2.4.2 and a new section in Appendix E.2), i.e. sufficient statistics also has the benefits of SimCLR. Additionally, all experiments show sufficient statistics with $M>2$ outperforms SimCLR ($M=2$). What we do observe is that PVC-Geometric outperforms sufficient statistics, but the reason cannot be due to SimCLR for the above reasons and is an interesting direction for future work.
> >
> > Again, we appreciate the time you took reading our work and sharing your valuable comments and questions. We will continue to clarify the manuscript according to the above. These enhancements will contribute to the overall readability and clarity.

---

> ### Author Response · Authors · 2023-11-20
> **Request for remaining issues**
>
> Dear Reviewer AhEA,
>
> We thank you again for your thoughtful comments and questions. The resulting changes have increased the completeness and readability of the work.
>
> We are still working to produce the requested fine-tuning results and will post them upon completion.
>
> Please let us know if you have any other issues remaining with the work.
>
> Best,
>
> The Authors

---

> ### Author Response · Authors · 2023-11-21
> **Requested results for fine-tuning on transfer tasks**
>
> Dear Reviewer AhEA,
>
> We are happy to let you know that the majority of our fine-tuning transfer evaluations are complete and have been added to Appendix F.2.2.
>
> For equivalent hyperparameters, we compared SimCLR against Growing Batch and Shrinking Batch variations of Geometric PVC. We did this for both small and large amounts of pre-training epochs, then fine tuning on a suite of natural tasks.
>
> On the majority of natural tasks, Geometric PVC significantly outperforms SimCLR, and for the remaining tasks they are statistically equivalent according to standard statistical tests. This demonstrates the utility of Poly-view methods on real tasks, addressing the outstanding W4/Q1. Thank you again for suggesting this investigation to us.
>
> Please let us know if you have any other issues remaining with the work.
>
> Best,
>
> The Authors

---

### Official Review · Reviewer_ymcZ · 2023-11-03

**Soundness:** 3 good
**Presentation:** 3 good
**Contribution:** 3 good
**Rating:** 8
**Confidence:** 4

**Summary:**

The paper proposed a multi-view (against the previous 2-view) contrastive learning, they provide theoretical and empirical evidence that their derived multi-view loss is better than previous multi-crop loss, as it provides a tighter lower bound on the generalized mutual information. They also provide real data evidence showing that their multi-view loss allows more efficient learning compared with previous two-view contrastive learning like SimCLR.

**Strengths:**

The paper investigated an interesting angle of contrastive learning: instead of increasing batch size, they increase the number of views. They provide a solid theoretical framework for their proposal, linking their proposed multi-view loss with previous SimCLR loss and the InfoMax framework. They also provide detailed analysis for comparing these two losses both theoretically and empirically. Overall it is clearly written and easy to follow, and the theoretical analysis aligns with the empirical findings is another big plus. Overall, they provide a new angle to improve the contrastive learning idea, which I believe might unleash further power of self-supervised learning.

**Weaknesses:**

I just have one suggestion: maybe you can comment (or leave for future work) about how other self-supervised learning can fit in your framework, or how your idea can be extended to other SSL methods like BYOL etc.

**Questions:**

I have no questions.

---

> ### Author Response · Authors · 2023-11-16
>
> Thank you for taking the time to read our work. We are happy you appreciate the empirical and theoretical validation of the Poly-View contrastive methods, their practical benefits as measured by efficiency, and recognize the broad applicability for improving self-supervised learning.
>
> Please find below our answer to your raised question.
>
> ### *Q1: I just have one suggestion: maybe you can comment (or leave for future work) about how other self-supervised learning can fit in your framework, or how your idea can be extended to other SSL methods like BYOL etc.*
>
> Our primary contributions use the frameworks of information theory and sufficient statistics to investigate what is possible in the presence of a view multiplicity $M>2$, and derive the different Poly-View objectives from first principles.
>
> It is possible to incorporate multiplicity $M>2$ into a distillation setup like BYOL (https://arxiv.org/abs/2006.07733). For example, DINOv1 (https://arxiv.org/abs/2104.14294), which shares many algorithmic parts of BYOL, benefits a lot from using the pair-wise Multi-Crop task that we describe in Section 2.2 and Appendix D (although in DINOv1, there is more than one augmentation policy).
>
> One option for extending distillation methods like BYOL/DINOv1 from Multi-Crop to Poly-View type tasks in a One-vs-Rest sense is to have the EMA teacher produce $M-1$ logits, which are aggregated into a single logit (similar to the sufficient statistics choice for $Q$ in Equation 24) for producing the target pseudo-label distribution.
> The gradient-based student could then be updated based on its predictions from the held-out view, and this procedure aggregated over all the view hold-outs.
>
> Such a procedure is very interesting and definitely worth future consideration.
> The core difference between the distillation procedure above and the Poly-View contrastive methods is that the large-view limit of Poly-View contrastive methods is provably a proxy for InfoMax (Section 2.1 and Equation 27). There may be a way to obtain theoretical guarantees for the large-view distillation methods (using for example tools from https://arxiv.org/abs/2307.10907), and could prove an interesting future direction for investigation.
>
> We have included a discussion regarding extensions of other self-supervised learning methods in Appendix H. Again, we appreciate the time you took to read our work and share your valuable comments and enthusiasm.

---

### Author Response · Authors · 2023-11-16
**Summary of revisions**

We have uploaded a revision based on feedback and questions from reviewers.

Summary of revisions:
1. [Page 5] Clarification in equations 12 and 13 which pieces correspond to PVC loss functions
2. [Page 6] Clarification that the exponential family form of sufficient statistics follows from the finite-dimensional nature of the representations, and is *not* an assumption.
3. [Page 7] Signposting for new derivation of $M=2$ case of sufficient statistics.
4. [Page 9] Signposting for a FLOPs comparison of Poly-View against pair-wise methods.
5. [Page 28] New derivation of $M=2$ case of sufficient statistics recovering SimCLR.
6. [Page 31] New proof of existence of finite optimal multiplicity $M^*$ in the *Fixed Batch* setting.
7. [Page 35] Fine-tuning results on transfer tasks demonstrating utility of Poly-view methods for transfer.
8. [Page 37] Total FLOPs comparison of Poly-View against pair-wise methods.
9. [Page 41] New discussion on incorporating Poly-View type computations in other types of self-supervised learning methods.

All modifications are highlighted in green for convenience.

Many thanks,

The authors

*Update: the list above has been updated to accommodate the inclusion of 7. transfer learning results and any resulting page number changes.*

---

### Meta-Review · Area_Chair_fASm · 2023-12-05

**Metareview:**

This paper extends the idea of contrastive learning to use more than two related reviews. The paper derived a new objective for the poly-view setting and demonstrated that contrastive models trained this way outperform traditional approach. Overall, the reviewers find the derivation of the new objective to be interesting and the empirical results are convincing.

**Justification For Why Not Higher Score:**

It's hard for me to decide where this should go, I can see it getting a spotlight but there might also be a lot of competition.

**Justification For Why Not Lower Score:**

All reviewers are positive on the paper and the idea seems interesting. The paper is also well-written.

---

### Decision · Program_Chairs · 2024-01-16

Accept (poster)